# The Epithelial Sodium Channel—An Underestimated Drug Target

**DOI:** 10.3390/ijms24097775

**Published:** 2023-04-24

**Authors:** Rosa Lemmens-Gruber, Susan Tzotzos

**Affiliations:** 1Department of Pharmaceutical Sciences, Division of Pharmacology and Toxicology, University of Vienna, A-1090 Vienna, Austria; 2Independent Researcher, A-1180 Vienna, Austria; stzotzos@mac.com

**Keywords:** epithelial sodium channel, gain- and loss-of-function mutations, Liddle syndrome, cystic fibrosis, pseudohypoaldosteronism type 1B, TIP peptides, solnatide, setanaxib

## Abstract

Epithelial sodium channels (ENaC) are part of a complex network of interacting biochemical pathways and as such are involved in several disease states. Dependent on site and type of mutation, gain- or loss-of-function generated symptoms occur which span from asymptomatic to life-threatening disorders such as Liddle syndrome, cystic fibrosis or generalized pseudohypoaldosteronism type 1. Variants of ENaC which are implicated in disease assist further understanding of their molecular mechanisms in order to create models for specific pharmacological targeting. Identification and characterization of ENaC modifiers not only furthers our basic understanding of how these regulatory processes interact, but also enables discovery of new therapeutic targets for the disease conditions caused by ENaC dysfunction. Numerous test compounds have revealed encouraging results in vitro and in animal models but less in clinical settings. The EMA- and FDA-designated orphan drug solnatide is currently being tested in phase 2 clinical trials in the setting of acute respiratory distress syndrome, and the NOX1/ NOX4 inhibitor setanaxib is undergoing clinical phase 2 and 3 trials for therapy of primary biliary cholangitis, liver stiffness, and carcinoma. The established ENaC blocker amiloride is mainly used as an add-on drug in the therapy of resistant hypertension and is being studied in ongoing clinical phase 3 and 4 trials for special applications. This review focuses on discussing some recent developments in the search for novel therapeutic agents.

## 1. Introduction

The critical role of the epithelial sodium channel (ENaC) in the maintenance of blood pressure, electrolyte and fluid balance is borne out by the occurrence of disease conditions in which ENaC function is disrupted by mutation. Owing to its role in sodium reabsorption and hence regulation of salt and water homeostasis, ENaC function is vital for human health. Therefore, identifying and characterizing modifiers of ENaC not only furthers our basic understanding of how these regulatory processes interact in the physiological state, but also enables discovery of new therapeutic targets for the disease conditions caused by ENaC dysfunction. Mutations in ENaC generating gain- or loss-of-function, can cause hereditary diseases such as Liddle syndrome (LS), cystic fibrosis (CF) or generalized pseudohypoaldosteronism type 1 (PHA-1B), which occur in different phenotypes. Thus, ENaC is an interesting target in drug discovery to develop new drug candidates for therapeutical use especially in salt-sensitive hypertension (HTN), CF and PHA-1B.

## 2. Phylogeny

ENaC belongs to the ENaC/degenerin (ENaC/DEG) family of cation-selective ion channels [1,2,3] which spans a wide phylogenetic spectrum and is associated with functions related to sensing and responding to the cell environment, mechanical and chemical stimuli [3,4,5]. Compared with its family members, ENaC is highly Na(+) selective (100 Na(+): 1 K(+)) and amiloride sensitive (EC_50_ of 150 nM). Unlike most ENaC/DEG family members which are quiescent at rest [3,4,5,6], ENaC has evolved as a constitutively active channel. This property enables the bulk transport of Na(+) across epithelia where ENaC activity is regulated by both intracellular and extracellular factors, as well as intramembrane lipids that modulate channel open probability [7]. ENaC/DEG proteins and constituent subunits are characterized by having a relatively long extracellular loop bounded by two transmembrane helices (TM1 and TM2), with shorter N-terminal and C-terminal regions located intracellularly. TM1 and TM2 form the channel’s membrane pore, which is the location of the channel’s gate [7]. Other members of this family include the DEGs that are part of mechanotransduction complexes in *Caenorhabditis elegans* [8], the peptide-gated channel Phe-Met-Arg-Phe-amide (FMRFa)-activated Na(+) channel (FaNaC) of snails [9], the mammalian bile acid-sensitive ion channel (BASIC) [10] and the Drosophila ENaC/DEG channels such as pickpocket, ripped pocket and others [11]. Amino acid sequence identity between different ENaC/DEG subfamilies is 15–20% [6].

## 3. Tissues in Which ENaC Is Expressed

ENaC is expressed in epithelial and non-epithelial cells in various tissues and organs throughout the body [12,13,14] as well as over the entire length of multi-ciliated cells in the lung and reproductive tract [15]. In the colon and the apical membrane of principal cells (PCs) of the aldosterone-sensitive distal nephron (ASDN) of the kidney, ENaC is highly expressed, mediating Na(+) entry from the lumen to the cell and therefore is crucial for the maintenance of blood Na(+) and K(+) levels and their homeostasis [3,5,7,16]. ENaCs in lingual epithelia participate in salt taste and influence Na(+) ingestion [17]. In the lung, Na(+) transport through apically-located ENaC in the alveolar epithelium is essential for maintaining the correct composition and volume of alveolar lining fluid, enabling optimal gas exchange [18]. ENaC in endothelial and vascular smooth muscle cells [19,20,21,22] has been shown to determine vasoconstriction by negatively modulating nitric oxide release in mesenteric arteries [23]. Increased ENaC expression could cause stiffening in endothelia resulting in reduced availability of nitric oxide—a characteristic early sign of HTN [20]. ENaC is an important regulator of vascular nanomechanics and a transducer of mechanical forces and so potentially influences the onset of endothelial dysfunction and HTN [21,24,25]. Furthermore, ENaC has been detected in neurons [26], blood cells [27,28,29], bone [30,31] and in the reproductive tract, namely in the endometrial [32] and germ cells [33,34]. Keeping this in mind, the availability of well-characterized, subunit-specific antibodies is mandatory for studying the role of ENaC in physiological and pathophysiological conditions as channel activity, subunit composition and expression levels in non-epithelial tissues vary and may differ from epithelial cells [35].

## 4. Structure and Stoichiometry

The first structure of the ENaC/DEG family to be resolved was that of the acid-sensing ion channel ASIC1, which is expressed in vertebrate central and peripheral nervous systems and plays a role in nociception and mechanosensation [36]. The 3D structure of ASIC1, a homotrimer, gave important clues as to the extracellular structure and pore regions of ENaC, providing insight into channel assembly, processing and how ENaC interacts with the external environment [37]. Recently, the structure of human ENaC (hENaC) has been solved by cryo-electron microscopy (cryo-EM) and has confirmed the prediction of features based on the ASIC1 structure [38]. Unlike homotrimeric ASIC1, ENaC is a heterotrimer of homologous subunits, α, β and γ, with a stoichiometry of 1:1:1 arranged in counterclockwise fashion (α, γ, β) when viewed from above the cell surface [38]. A more recent structure at 3 Ångstroms resolution has revealed that homotrimers of α-ENaC are possible but that homotrimers of β and γ are sterically unfavourable [39]. Thus, the subunit stoichiometry is conserved within the ENaC/DEG family. A fourth subunit, δ, has been observed in human tissue of brain, eye, heart, liver, lung, ovaries, pancreas and testes, and can substitute for α-ENaC, forming δβγ-ENaC, which has different properties from the more frequently observed, αβγ-ENaC [13,20,21,40,41,42]. Hitherto, δ-ENaC has been found in some rodents except mice and rats [12]. Future investigations of the potentially important role of δ-ENaC in human vascular physiology and pathology require new translational models instead of existing rodent ones [21].

Prior to the resolution of the cryo-EM structure for ENaC, early evidence from functional and biochemical studies had suggested that ENaCs were heterotetramers comprising two α-, one β- and one γ-subunit [43,44,45]. Evidence also exists for ENaC-like channels with an alternative subunit composition from the preferred αβγ-ENaC. The non-selective sodium ion channels (NSC) which contribute to alveolar fluid clearance and have been described as being relatively non-selective for Na(+) over K(+), having a larger conductance, and shorter mean open and closed times than the typical highly selective ENaC channels (HSC), are believed to be trimers of one α-ENaC with two ASIC1a subunits [46]. Electrophysiological studies in heterologous expression systems of hENaC subunits, have indicated that channels comprising only α-subunits and only δ-subunits are functional and respond to activation by synthetic peptides, although current amplitudes are significantly less than those observed for αβγ-hENaC or δβγ-hENaC [47].

The large extracellular loops of ASIC1 and ENaC show clearly defined domains termed finger, thumb, palm, knuckle and β-ball owing to their overall arrangement resembling a raised hand holding a ball; the wrist lies where the lower palm and base of thumb converge with the transmembrane domain [36,38]. The domains differ according to their α-helical and β-strand content. Notably, the degree of sequence conservation between homologous ENaC subunits and ASIC1 varies among these defined extracellular domains. Sequence identity between ENaC subunits and ASIC1 is highest (33–36%) in the palm and β-ball domains that form the inner core of the assembled channel complex and adopt a β-sheet secondary structure. Sequence identity is much lower in the peripheral thumb, knuckle, and finger domains that are seemingly modular and characterized by higher α-helical content [37]. Researchers have speculated that these poorly conserved regions play key roles in conferring specificity regarding the parameters that regulate distinct members of the ENaC/DEG family [48]. The selective activation of ENaC by proteases that target unique regions in the extracellular domains of the α- and γ-subunits is a good illustration of this phenomenon [37,38].

Before the structure for hENaC had been resolved [38], the low sequence identity within the finger domain between ENaC subunits and ASIC1, as well as the large inserts (70–87 residues) in the ENaC finger domains, hampered using the resolved ASIC1 finger domain structure to generate models of ENaC finger domains. The structure for hENaC has illustrated the critical divergence of ENaC from ASIC1 in the peripheral region of the extracellular domain. ENaC differs significantly from ASIC1 in both structure and primary sequence of the knuckle and finger domains [38].

## 5. Structural and Biochemical Regulation of ENaC Function

ENaC is regulated by a plethora of extracellular and intracellular metabolites, some of which interact via the sequence motifs outlined below. Hormones such as angiotensin (Ang II), aldosterone, vasopressin, glucocorticoids and mineralocorticoids play major roles in ENaC regulation. Ions, such as Na(+), K(+), Li(+), Zn(2+), Cl(−), F(−), phospholipids (e.g., PIP2 and PIP3), several proteins (e.g., NEDD4-2, SGK1, proteases, kinases, TNF, endothelin-1 and others) and peptides interact with ENaC directly modifying activity of the channel.

Recently, proteomic studies in kidney epithelial cells revealed that sodium fluoride is involved in signal transduction. Sodium fluoride reduced cell viability in a concentration dependent manner. At moderate concentrations of 100 and 200 µM sodium fluoride, ENaC subunit genes *SCNN1A* and *SCNN1G*, but not *SCNN1B*, were up-regulated, whereas at a high concentration of 400 μM all three ENaC subunit genes were down-regulated [49].

Posttranslational modifications such as ubiquitination, phosphorylation, acetylation and palmitoylation subtly fine-tune ENaC activity to that required by the cellular environment and needs of the organism as a whole. As mentioned earlier, an essential response of ENaC is that towards shear stress which plays a critical role in the regulation of blood pressure [50]. ENaC is part of a complex network of interacting biochemical pathways and as such is implicated in several disease states brought about by their dysfunction.

The reader is referred to recent reviews for a more detailed description of the topic: [7,13,16,20,23,48,51,52,53,54,55,56].

### 5.1. Regulation of ENaC Activity by Proteases

As mentioned earlier, ENaC activity is primarily modulated by proteases that remove peptidyl tracts in the extracellular domain (ECD) [6,57,58,59]. Removal of these inhibitory peptides irreversibly converts ENaC channels from a low to a high open probability (Po) state [60,61]. Activation of ENaC is required for Na(+) reabsorption across epithelia and so any dysregulation may result in disease, such as CF, salt-sensitive HTN, nephrotic syndrome or PHA-1.

Canonically, the α-subunit is cleaved twice by furin, while the γ-subunit is cleaved once by furin and once by the channel activating protease 1 (CAP1), also called serine protease 8 (PRSS8) or more usually prostasin [60,61,62,63,64]. The β-subunit does not have canonical protease sites. The structure of ENaC has provided essential insight into the positions and molecular environment of these protease-sensitive regions, enabling the characterization of the “Gating Release of Inhibition” (GRIP) domains [38]. The GRIP domains are stretches of 20–40 amino acid residues which lie at the periphery of the heterotrimeric structure, and in the primary sequence are located between the α1 and α2 helices of the finger domains in each subunit. The GRIP domain encompasses inhibitory peptides and flanking protease sensitive sites, which are arranged topologically adjacent to residues in the thumb domain of the same subunit, such that the GRIP domain forms a wedge between the finger and thumb regions [38]. The β-ENaC subunit also contains a GRIP domain, even though it is not cleaved by proteases.

The earlier research determining the role of proteases in ENaC activation was mostly carried out in mice and rats in vivo and in heterologous expression systems. Validation that these processes also occur in human kidney tissue in diseased and healthy states has recently been obtained [65]. Nevertheless, understanding of the mechanisms of protease activation of ENaC in the context of healthy and disease states in humans remains incomplete [16,48], and the precise mechanism of ENaC modulation by proteases has not been fully elucidated [66]. The current paradigm of protease activation of ENaC is that three cleavage sites (two in α-ENaC and one in γ-ENaC) are targeted by furin and/or related furin-like proprotein convertases during channel maturation in the intracellular biosynthetic pathway [48,66,67,68]. The pivotal final step of proteolytic ENaC activation is assumed to take place at the plasma membrane where γ-ENaC is cleaved by membrane-anchored and/or extracellular proteases in a region distal to the furin site [68]. Apart from prostasin, several other candidate proteases which cleave the γ-subunit, have been implicated in the final step of proteolytic ENaC activation, amongst them factor VII activating protease (FSAP) [69] and transmembrane serine protease 2 (TMPRSS2) [68] (Figure 1). The reader is referred to a recent review by Anand and co-researchers for a comprehensive and up-to-date account of ENaC activation by proteases [66].

A striking finding during the recent SARS-CoV-2 pandemic was that the furin site of the spike protein (S) of the virus, the cleavage of which results in the separation of the two subunits of the protein, S1 and S2, is identical to the furin site spanning residues R201-R204 (Figure 1) in human α-ENaC [70]. Viruses hijack the host cell’s biochemical toolbox, making use of host enzymes to enter the host cell and reproduce themselves at the host’s expense. It has been hypothesized that by diverting furin from the cellular membrane environment, SARS-CoV-2 could compromise ENaC function with the consequences of underactive ENaC contributing to COVID-19 symptoms [71].

**Figure 1 ijms-24-07775-f001:**
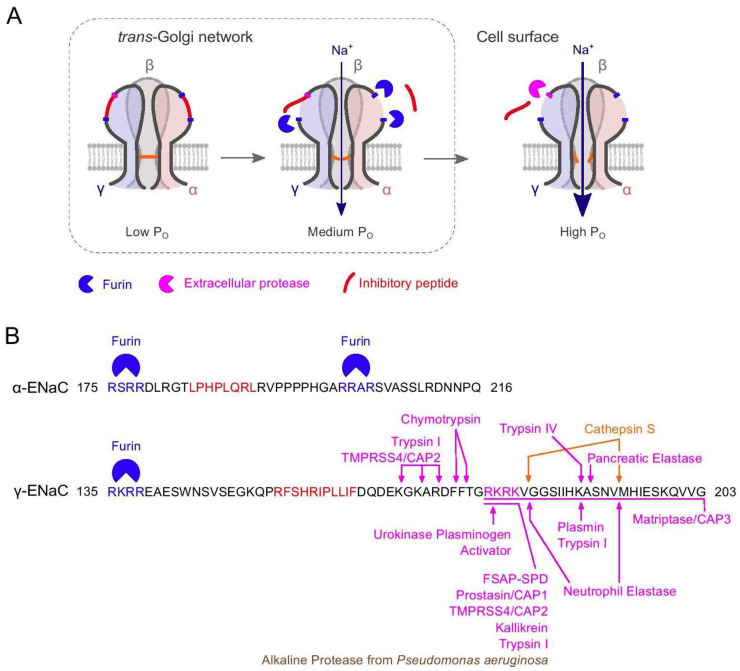
ENaC activation by intra- and extracellular proteases. (A) Cartoon illustration of ENaC cleavage by furin within the *trans*-Golgi network and extracellular proteases at the cell surface. P_O_, open probability. (B) Peptide sequences of the α- and γ-subunits of human ENaC showing the regions containing the inhibitory peptides within the extracellular loop. Furin consensus sites are shown in blue; the inhibitory peptides [38] are shown in red. The figure shows extracellular proteases cleaving the γ-subunit whose cleavage sites have been identified by mutagenesis studies [48,72,73]: Serine proteases are shown in magenta, cysteine protease in orange, and metalloprotease in brown. CAP, channel-activating protease (alternative name for the indicated proteases); FSAP-SPD, factor VII activating protease—serine protease domain. Reproduced from [74] (Figure 1); Copyright © The Authors, 2021; published by Springer, Berlin, New York.

### 5.2. Sequence Motifs Involved in Regulation of ENaC

In addition to the protease cleavage sites associated with the removal of inhibitory tracts in α- or γ-ENaC which activates/opens the channel, the large ECDs of the ENaC subunits contain other conserved motifs and sites which play critical roles in the regulation of channel activity [13]. The cytoplasmic N- and C-terminal regions of the channel subunits serve as binding sites for several important regulatory proteins, and for posttranslational modifications, such as ubiquitination, phosphorylation, acetylation, and palmitoylation. These also include sites that are mutated in human diseases such as LS and PHA-1B (below).

#### 5.2.1. Selectivity Filter

Within the TM2 region, the ion selectivity filter of ENaC, consisting of the conserved amino acid motif G/SxS, is located at the centre of the TM2 region in the hENaC structure. The ENaC selectivity filter facilitates selective passage of Na(+) compared to K(+) ions and is at least 10-fold more selective than the ASIC channels [38].

#### 5.2.2. Amiloride-Binding Site

The residue S556 in human α-ENaC, and equivalent glycine residues in β- and γ-subunits, which lie in the middle of TM2 and are conserved in mouse and rat ENaC, are involved in amiloride-binding. Mutation at this position in all three subunits removes any sensitivity to amiloride blocking [75]. A recent study of the cryo-EM structure of hENaC [38,39] by THz spectroscopy, coupled with molecular modelling, confirms this as the amiloride binding site [76].

#### 5.2.3. N-Glycosylation

ENaCs are glycosylated at between 5 and 12 asparagine (Asn) residues per subunit [77,78]. N-linked glycans on each subunit are required for maturation, proper folding, surface expression, and function of the channel [79]. Glycan addition is regulated by sodium and affects protease activation at the cell surface, protein trafficking, sodium-dependent regulation and sodium transport. Glycosylation of the α-subunit also determines whether a chaperone, Lhs1/GRP170, selects the protein for endoplasmic reticulum-associated degradation. Recognition by this chaperone is blocked by the assembly of the ENaC transmembrane domains [51]. N-glycosylation has been shown to be essential for the interaction of ENaC with peptides known to enhance the Po of the channel [47]. Glycosylated Asn residues in the palm and knuckle domains of α-ENaC have been implicated in the ENaC shear force response and identified as potential tethers [50]. The role of N-glycans attached to ENaC acting as tethers for mechanotransduction has been further supported by work showing that insertion of N-glycosylation motifs increases the shear force responsiveness of δ-ENaC [80].

#### 5.2.4. Cysteine-Rich Domains

The most notable feature of the extracellular loop of the ENaC/DEG family of ion channels is the presence of two highly conserved cysteine-rich domains (CRD1 and CRD2) covering about 50% of the sequence [81]. Their high conservation suggests that the cysteine residues are involved in disulfide bond formation. About half the Cys residues on the ECL of ENaC are essential for maintaining the scaffold necessary for Na(+) self-inhibition [82]. In the recently solved cryo-EM structure of hENaC, eight disulfide bridges in the extracellular loops of α- and γ-subunits, and nine in the β-subunit, stabilise the architecture resembling a hand with palm, knuckle, finger, and thumb domains clenching a ‘ball’ of β-strands [38]. Seven of the disulfide bonds are conserved throughout the ENaC/DEG family, the eighth is unique to the three ENaC subunits [38]. Cysteine-rich domains (CRD) are involved in intrasubunit interactions between the finger and thumb domains within the α- or γ-ENaC subunits, thus regulating channel function (Figure 2) [83]. The importance of the wrist region and the critical roles that disulfide bridges play in maintaining the structural and functional integrity of ENaC is emphasized by alterations of a conserved cysteine, α-Cys479 to an Arg, causing LS. This missense mutation not only eliminates a disulfide bridge located at the juncture of the thumb and palm domains but also introduces a bulky, positively charged residue [84].

#### 5.2.5. Na(+)-Binding Acidic Cleft

A Na(+)-binding site has been detected in a conserved acidic pocket in the periphery of the extracellular loop of α-ENaC which is involved in the inhibitory response of the channel to high extracellular Na(+) concentrations [39,85]. Extracellular Na(+) binds to ENaC at this site, driving allosteric changes that are transmitted to the channel gate, reducing ENaC Po [7,85]. A His residue in γ-ENaC has been implicated in this response [86]. This response, referred to as Na(+) self-inhibition [79], provides a mechanism for cells in the distal nephron to rapidly adjust the rate of Na(+) influx according to changes in the concentration of urinary Na(+) [48]. To date, over 90 amino acid missense mutations resulting in altered channel activity, primarily due to altered Na(+) self-inhibition, have been observed at various sites in the extracellular regions of human and mouse ENaC [5,58,82,86,87,88,89,90,91]. Most of these mutations likely affect allosteric transitions that occur in response to Na(+) binding to the channel [7,48].

ASIC and ENaC subunits have a localized extracellular cluster of acidic residues which in the ASIC structure span strand β6 in the β-ball, and strand β7 in the palm domains, and that could facilitate Na(+) binding [5,36,85]. In the ENaC architecture, strands β6 and β7 form a loop, which lies at the juncture of the finger, β-ball and thumb domains and adjacent to the α1 helix in the finger domain [38,39]. In heterologous expression studies of mutants of mouse α-ENaC generated by Trp scanning mutagenesis, the effects of individual mutations in the region of acidic cleft Asp and Glu residues on Na(+) self-inhibition were observed [85]. Mutant channels showed a modified Na(+) inhibitory response as well as an altered cation selectivity which varied depending on the residue involved, but the overall conclusion was that residues within the acidic pocket form an allosteric effector binding site for Na(+). It was also found that crosslinking the β6-β7 loop to the α1 helix of the finger domain reduced channel activity [85].

#### 5.2.6. HG-Motif

The intracellular N-terminal region is important for regulating ENaC gating, particularly the HGxxR sequence that is conserved in all three subunits [92,93]. A very recent cryoEM structure of the ENaC relative ASIC1 reveals that this HG motif is part of a re-entrant loop that lines the permeation pathway [94]. This possibly has implications for the regulation of gating and ion selectivity ofENaC. Disruption of the HG motif in ENaC leads to a lower Po of ENaC and is found in some patients suffering from generalized pseudohypoaldosteronism, PHA-1B, in whom ENaC activity is reduced compared to normal [92] (see below).

#### 5.2.7. N-Terminal Ubiquitination Motif

The N-terminal region of α- and γ-ENaC contains several clusters of Lys residues that are ubiquitinated by neural precursor cell expressed developmentally downregulated protein (NEDD4-2) (see below), leading to ENaC endocytosis and degradation [95]. These are the major sites modified by ubiquitination following binding of the NEDD4-2 ubiquitin-protein ligase to PY motifs in the C-terminal region of ENaC α-, β- and γ-subunits [96]. Ubiquitination affects the trafficking of ENaC protein within the cell, including the rates of internalization from the cell surface and proteasomal degradation [16]. Ubiquitination occurs after complete proteolytic processing of the subunits, contributing to retrieval and/or disposal of channels expressed at the cell surface [96]. The mechanisms by which Nedd4-2-dependent ubiquitination promotes ENaC internalization are not yet fully understood, but probably include clathrin-mediated endocytosis [97]. Epsin, which has both ubiquitin-interacting and clathrin-binding motifs, may serve as a link between ubiquitinated ENaC subunits and clathrin-dependent endocytic elements [98].

In post-translational modification of ENaC, the ubiquitin ligase NEDD4 plays an important role through ubiquitination, which regulates ENaC expression on the cell membrane. ENaC retrieval from the cell surface links with the interaction of NEDD4-2 with proline-rich PPPxY-motifs in the C-terminal region of ENaC subunits. This interaction occurs via WW-domains of NEDD4-2, promoting ubiquitination, endocytosis, and proteasomal degradation of the channel. The role of E3-ubiquitin ligase NEDD4L/NEDD4-2 as regulators of ENaC in aldosterone-sensitive distal nephrons has been extensively reviewed by Ishigami et al. [52]. Frindt et al. [96] reported that ubiquitinated α-ENaC was detected in tissue homogenates of mouse and rat kidney with variable numbers of ubiquitin molecules primarily at the N-terminal cleaved fragment of the α-subunit. However, no significant ubiquitination of β-ENaC was detected. For γ-ENaC, stable attachment of the ubiquitinated N-terminus to the C-terminal moiety was indicated. In mice with the LS mutation β566X, deletion of a putative binding site for the ubiquitin ligase NEDD4-2 is evident, leading to a reduction of ubiquitinated β-ENaC. Sodium depletion increased the quantities of ubiquitinated α- and γ-ENaC. It is concluded that ubiquitination occurs after complete proteolytic processing of the subunits. Ubiquitinated channels are rapidly internalized and degraded fairly independently of the regulation by aldosterone, with the rate of ubiquitination depending on the number of channels at the surface. Tight regulation of active channels at the surface is thus achieved by limiting their lifetime [96]. 

#### 5.2.8. PY Motif

The cytoplasmic C-terminal regions of each of the α-, β-, γ- ENaC subunits contain a short sequence (PPPxY) called the PY motif, which serves as a binding site for the WW domains of the NEDD4-2 ubiquitin-protein ligase [99,100,101,102,103]. Mutations associated with LS disrupt the PY motif in the C-terminal region of the β- or γ-subunit of ENaC, which impairs interaction of the channel with the ubiquitin ligase Nedd4-2, resulting in an increase in the expression of channels at the plasma membrane as well as an increase in channel Po [48]. Overlapping the PY motif in each of the three ENaC subunits is a putative tyrosine-based (Y*XX*ϕ, where ϕ represents a residue with a hydrophobic side chain) endocytic/trafficking signal [104].

#### 5.2.9. Cys-Palmitoylation

The β- and γ-subunits, but not the α-subunit, of both mouse and hENaC are palmitoylated in their cytoplasmic termini [105,106,107]. The β-subunit of mouse ENaC contains two palmitoylation sites, one in the N-terminal region at C43 and one in the C-terminal region at C557 (A559 in human β-ENaC is equivalent to C557 in mouse β-ENaC). The β-subunit of hENaC has only a cysteine, C43, in the N-terminal region. The γ-subunit of human and mouse ENaC contains two palmitoylation sites both situated in the N-terminal cytoplasmic domain at C33 and C41. Previous work has indicated that the palmitoylation sites within the β- and γ-subunit N-terminal regions are in a location important for the regulation of channel gating. The sites are adjacent to the highly conserved HG-motif (mentioned previously) present in the α-, β- and γ-subunits of human and mouse ENaC. Mutation in the HG-motif of the β-subunit has been shown to result in a loss-of-function ENaC and a PHA-1-like phenotype in the homozygous state [92]. Cys-palmitoylation of ENaC appears to stabilize the open state of the channel, and ENaC activity is reduced by blocking its modification by palmitoylation [7]. The structure of hENaC unfortunately does not contain details of the cytoplasmic domains [38,39]. However, the secondary structure of the cytoplasmic tails of the β- and γ-subunits has been predicted using prediction algorithms in order to give some insight into how Cys-palmitoylation might affect pore structure in the TM domains [105,107]. The N-terminal cytoplasmic domains contain two predicted α-helices and the C-terminal domain a single α-helix. The N-terminal palmitoylation site in the β-subunit lies 7 residues from TM1 in a loop between TM1 and one α-helix. The palmitoylation site in the C-terminal of the β-subunit (of mouse ENaC) lies on a hydrophobic face of the α-helix, a position where it would stabilize the interaction of the helical region with the plasma membrane and influence the angle of the adjacent TM2 domain. The predicted structure of the γ-subunit N-terminal cytoplasmic tail places one palmitoylation site at the terminus of one α-helix and the second site in the adjacent loop, 12 residues from the TM1 domain. The location of these palmitate residues on the β- and γ-subunit N-terminal domains is likely to influence the channel pore and gate by altering the disposition of TM1 [7].

#### 5.2.10. Acetylation

Cytosolic lysines in ENaC subunits are substrates for acetylation, carried out by histone acetyltransferases which catalyze transfer of an acetyl group from acetyl-CoA to the target lysine. Acetylation of lysines inhibits their ubiquitination and consequently the rate of endocytosis of channels at the cell surface. In antagonizing ENaC ubiquitination, acetylation increases epithelial Na(+) absorption [108].

#### 5.2.11. Phosphorylation

Specific sites in the C-terminal cytosolic regions of ENaC are susceptible to phosphorylation by kinases, which modify the interaction of the channel with other regulatory proteins. The G protein-coupled receptor kinase 2 (Grk2), acts on S633 in mouse β-ENaC (equivalent to S635 in human β-ENaC) and this phosphorylation renders the channel insensitive to binding Nedd4-2 and ensuing ubiquitination [109]. In this case phosphorylation enhances ENaC activity by bringing about an increase of channel abundance at the cell surface. Kinases can also stimulate or inhibit ENaC activity directly by increasing channel Po, as has been recently demonstrated for tyrosine-phosphorylation-regulated kinase 2 (DYRK2) and glycogen synthase kinase 3 β (GSK3β) in electrophysiological experiments using *Xenopus* oocytes heterologously expressing rat ENaC [110]. The typical phosphorylation recognition sequence of DYRK2 is RXX(S/T)P, which is found highly conserved in the C-terminal region of α-ENaC, close to TM2 [110] (Figure 3). It was shown that S621 in rat α-ENaC is phosphorylated by DYRK2 and that this phosphorylation activates ENaC. Furthermore, phosphorylation of S621 by DYRK2 primes the preceding S617 residue for phosphorylation by glycogen synthase kinase 3 β (GSK3β). However, phosphorylation by GSK3β results in channel inhibition. These researchers also showed that DYRK2 can also stimulate ENaC currents in microdissected mouse distal nephron, whereas GSK3β inhibits the currents [110]. In human α-ENaC the RXX(S/T)P phosphorylation motif spans R591–P595 with S594 being the predicted phosphorylation site for DYRK2 and S590 that for GSK3β (Figure 3).

Further, it has been demonstrated that Casein Kinase II-dependent phosphorylation at a site within a canonical “anchor” ankyrin binding motif activates ENaC and augments channel trafficking [111] which reveals another important mechanism in regulating ENaC-dependent Na(+) excretion. Consequently, regulation of ENaC by the Casein Kinase II/ankyrin-3 axis is hypothesized to be important for blood pressure control.

The metabolic sensor AMP-activated protein kinase (AMPK) inhibits ENaC via phosphorylation of β_1_Pix at Ser71. βPix (p21-activated kinase (PAK)-interacting exchange factor-β) is a member of the diffuse B cell lymphoma family of Rho guanine nucleotide exchange factors which binds to 14-3-3 proteins. The functional isoform β_1_Pix is required for ENaC inhibition by AMPK and promotes phosphorylation of NEDD4-2 to form an ENaC inhibitory β_1_Pix/NEDD4-2/14-3-3 complex in mouse kidney cortical collecting duct cells [112].

#### 5.2.12. PIP2 and PIP3 Binding Sites

Anionic phospholipids such as phosphatidylinositol 4,5-*bis*phosphate (PIP2) and phosphatidylinositol 3,4,5-triphosphate (PIP3) are found in the inner leaflet of the plasma membrane and play key roles in the regulation of ion transporters and channels. Two binding sites for PIP2 with a high concentration of basic residues have been identified in the cytosolic N-terminal region of β- and γ-ENaC [113,114,115]. One binding site for PIP3 has been identified in the N-terminal region of γ-ENaC [116]. Results of numerous studies suggest that the inositol phosphate head group is necessary for ENaC activation, and interaction with an inositol lipid phosphate is essential for ENaC to adopt the open state, reviewed in [48]. In electrophysiological experiments in cultured cells, PIP2 activates ENaC in a dose-dependent manner by increasing channel Po [115]. ENaC-PIP2 interaction occurs by electrostatic interaction, but is mediated by membrane-associated protein, myristoylated alanine-rich C kinase (MARCKS-like protein 1, MLP-1). MLP-1 interacts covalently with ENaC at a positively charged effector domain of MLP-1 near PIP2 binding sites in the cytosolic N-terminal region of β- and γ-ENaC [115], bringing about an increase in PIP2 concentration in the vicinity of ENaC, which is necessary for normal ENaC activity [117,118]. Native MLP-1 consists of several distinct structural elements. A positively charged effector domain sequesters PIP2, contains serines that are the target of protein kinase C (PKC), and controls MLP-1 association with the membrane. Additionally, MLP-1 has a myristoylation domain that promotes association with the membrane, and a multiple homology 2 domain. To further study the role of MLP-1 in distal convoluted tubule clonal cell line DCT-15 cells, Song et al. [114] constructed several MLP-1 mutants with the aim of preventing phosphorylation and myristoylation. They found that different mutants localize in different subcellular compartments depending on their preferred location in the membrane or in the cytosol, and that activation of PKC increases phosphorylation of endogenous MLP-1 and reduces ENaC activity [114].

### 5.3. Regulation of ENaC via Several Other Proteins

Cyclooxygenase-2/prostaglandin E2/E-prostanoid 1 signalling in the collecting duct site of the nephron has been identified as an important regulator of Na(+) homeostasis during Na(+) depletion. In cyclooxygenase-2 and E-prostanoid 1 deletion mouse models, higher urinary Na(+) excretion was observed as well as suppressed intrarenal renin, Ang II and aldosterone without affecting the systemic renin-angiotensin-aldosterone system (RAAS). Further, expression of the ENaC subunits in the deletion mouse model was reduced compared with the control mice. The direct effect of prostaglandin E2 was studied in the primary cultured inner medullary collecting duct cells. Exposure to prostaglandin E2 stimulated the release of soluble (pro)renin receptor, prorenin/renin and aldosterone. It was concluded that this signalling pathway might contribute to Na(+) homeostasis through activation of ENaC and the intrarenal RAAS [119].

Serum and glucocorticoid-regulated kinase 1 (SGK1) regulates several ion channels including ENaC (see [13]). SGK1 and ENaC in the luminal endometrium epithelium are critically involved in embryo implantation. When an endometrial adenocarcinoma model cell line of endometrial epithelial cells (Ishikawa cells) was treated with the negative regulator of uterine receptivity LEFTYA (aka endometrial bleeding-associated factor EBAF), the amiloride sensitive Na(+)-current was rapidly increased and expression of ENaC in the endometrium was induced. These effects could be diminished by application of an SGK1 inhibitor and were not observed in Sgk1-deficient mice. It is suggested that aberrant regulation of SGK1 and ENaC by LEFTYA could contribute to the pathogenesis of otherwise unexplained infertility [32].

Further, it was demonstrated that the transmembrane protein connexin 30 (Cx30), responsible for the formation of gap junctions between adjacent cells, is also involved in the regulation of ENaC. Cx30 promotes channel retrieval from the plasma membrane via clathrin-dependent endocytosis. Truncating the C-termini of β- or γ-ENaC significantly reduced the inhibitory effect of Cx30 on ENaC. Lack of this inhibition may contribute to increased ENaC activity. Due to reduced cell surface expression, Cx30 deficient mice show increased ENaC activity in the distal nephron [97].

Elevated formation of reactive oxygen species (ROS) leads to molecular damage. Redox signalling agents such as NADPH oxidases (NOX) play a role in physiology and disease. NOX activator 1 (NOXA1) is located in epithelial cells of the Henle’s thick ascending limb and distal nephron and mediates RAAS hyperactivation and ROS levels. Ang II increased NOXA1/NOX1 expression and induced ROS in the kidney of male wild-type mice by activating this enzyme, which caused enhanced tubular ENaC expression [120].

Paraoxonases are orthologs of *Caenorhabditis elegans* MEC-6, an endoplasmic reticulum-resident chaperone with an important function in proper assembly and surface expression of the touch-sensing degenerin channel in nematodes. MEC-6 and Paraoxonase 2 were shown to negatively regulate functional expression of ENaC. Paraoxonase 3 was found to be specifically expressed in the aldosterone-sensitive distal tubules of the mouse kidney, and to regulate ENaC expression by inhibiting its biogenesis and/or trafficking without affecting Po [121].

## 6. Variation in Human *SCNN1A*, *SCNN1B* and *SCNN1G*

Human genome sequencing projects continue to disclose an increasing number of ENaC gene variants. A comparison of the number of ENaC variants known a decade ago to the current estimate is a striking illustration of this. Thus, a survey of the single nucleotide polymorphism (SNP) database dbSNP (build 137, released June 2012 by the National Center for Biotechnology Information) [122] revealed 849, 2039, and 879 variations in human *SCNN1A*, *SCNN1B* and *SCNN1G*, respectively [123]; a similar survey conducted at the time of writing (February 2023) yielded 14,532 (*SCNN1A),* 48,227 (*SCNN1B*) and 16,124 (*SCNN1G*). Most of these occur in introns, but variants occurring in coding sequences and splice junctions which result in a change to protein sequence amount to 852 in human *SCNN1A*, 726 in *SCNN1B* and 575 in *SCNN1G*, respectively. Several of these have been found in individuals with salt-sensitive HTN, LS, CF, PHA-1B and other disorders [123,124,125,126,127,128]. According to one study, approximately 9% of individuals in the general population would be expected to carry a rare ENaC variant [125]. However, the majority of hENaC variants have not been characterized and their functional properties remain unknown [123,128].

### 6.1. Gain-of-Function: Liddle Syndrome

Liddle syndrome (LS) (or pseudoaldosteronism) (OMIM:177200) is an autosomal dominant form of salt-sensitive HTN associated with low plasma aldosterone, low plasma renin activity, hypokalemia and metabolic alkalosis. This severe form of salt-sensitive HTN is caused by gain-of-function mutations in the C-terminal region of the β- or γ-ENaC subunits [103,129]. Mutations in the last exon of the *SCNN1B* and *SCNN1G* genes delete the conserved proline-rich PY motif in the cytosolic C-terminal region of ENaC subunits [103,129], leading to an increase in the abundance and activity of ENaC at the cell surface [130,131]. Mutations causing LS disrupt the proline-rich PPPxY motif in the C-terminal region of β- and γ-ENaC subunits involved in specific interactions with cytosolic proteins, such as the Nedd4-2 ubiquitin ligase, that tightly control the channel density at the cell surface [132,133]. ENaC mutants lacking this proline-rich motif fail to undergo ubiquitination, and hence are not internalized, leading to retention of active channels at the cell surface [102,134].

Since these initial reports, a case of LS due to a gain-of-function mutation in the extracellular domain of the *α*-ENaC subunit (*SCNN1A*) that predominantly increases Po but not channel surface density has been observed [84]. Diagnostic exome sequencing revealed a novel heterozygous, nonconservative T > C single-nucleotide mutation in α-ENaC that results in the replacement of cysteine 479 with arginine (C479R). C479 lies in the second cysteine-rich domain (CRD2) of *α-*ENaC and is located at the beginning of strand β10 in the wrist of the ENaC structure at the base of the palm domain, forming a disulfide bridge with another highly conserved cysteine residue, C394. C479 is conserved not only amongst the ENaC subunits and ENaC homologs but also amongst ASIC1 orthologs [84], indicating the important structural role of this residue. Mapping of the C479 onto the cryo-EM structure of hENaC confirms the critical position occupied by this residue in the scaffold of α-ENaC. An earlier report of the gain-of-function alanine substitution at this site (C479A) which increases channel activity due to a loss of Na(+) self-inhibition [82], suggests that αC479R might also exhibit a loss of Na(+) self-inhibition [23]. Importantly, mutations of the cysteines in the extracellular cysteine-rich domains can result in either channel loss- or gain-of-function. Mutation of the human α-ENaC Cys133 into a tyrosine causes the mirror image of LS, generalized PHA-1 (PHA-1B), the severe salt-losing syndrome in neonates [135].

LS and PHA-1B remind us of the pivotal role played by ENaC in K(+) excretion: overactive ENaC in LS leads to hypokalemia; underactive ENaC in PHA-1B leads to hyperkalemia [16,136]. The occurrence of LS, albeit rare, emphasizes that minor disturbances of ENaC regulation resulting in increased ENaC activity are likely to contribute to the pathophysiology of essential hypertension, more so if these patients are genetically predisposed to salt-sensitive HTN [16].

### 6.2. Gain-of-Function: Cystic Fibrosis

CF (OMIM:219700) is a genetic disease caused by mutations in the cystic fibrosis transmembrane conductance regulator (CFTR) gene and is considered the most common severe monogenic disease inherited in an autosomal recessive fashion among populations of Western European ancestry. The CFTR is a cAMP-activated channel whose defective function in CF patients leads to abnormal Cl(−) and Na(+) transport in the sweat glands, respiratory tract, pancreas and male reproductive system [137]. In three independent CF patient populations of individuals homozygous for the most frequently observed CF-causing allele (F508del-CFTR), a variant of *SCNN1B* has been identified as having a modifying effect. The *SCNN1B* risk allele, rs2303153-C and the corresponding benign allele, rs2303153-G, were observed in all three patient cohorts [138] and an association was found between levels of expression of *SCNN1B* and the rs2303153 genotype in intestinal epithelia. In regulating β-ENaC expression, variant rs2303153 influences the course of CF disease. This confirms that ENaC is a CF modifier gene and as such is a potential therapeutic target for CF [138].

Amongst the gain-of-function mutations in hENaC, some reflect a loss of Na(+) inhibition [7,48]. Some of these have been discovered during genetic screening in patients diagnosed with CF. Increased activity of ENaC in the respiratory airways contributes to the pathophysiology of CF. In some patients suffering from atypical CF, a mutation can be identified in only one CFTR allele. The W493R gain-of-function mutant in human α-ENaC was found in an individual patient suffering from atypical heterozygous CF. The concurrence of the ENaC gain-of-function mutation with a mutation in one CFTR allele resulted in a CF phenotype in this individual [127]. The functional effects of this mutation were investigated by expressing wild-type αβγ-ENaC or mutant αW493Rβγ-ENaC in *Xenopus* oocytes. The αW493R mutation abolishes the Na(+) self-inhibition of ENaC, which contributes to its gain-of-function effects in the patient with atypical CF, who was also heterozygous for this mutation [127]. At the time of these experiments, prediction of the location of the conserved residue W493 in the α-ENaC architecture was only possible using a homology model of human α-ENaC based on ASIC1 structure. The W493 residue can be now accurately mapped to the available cryo-EM structure of hENaC, revealing its location in a short stretch of residues linking the β10 strand of the palm domain with the α6 helix of the knuckle domain, where it participates in an aromatic pocket interaction with residues in the γ-subunit [38].

Another gain-of-function variant in human γ-ENaC is L511Q (reference sequence ID: rs113234492) which is implicated in salt-sensitive HTN [123]. In a Xenopus oocyte expression system, αβγL511Q-ENaC showed more than fourfold amiloride-sensitive current than cells expressing wild-type hENaC and virtually eliminated the Na(+) self-inhibition response as well as protease activation (by chymotrypsin). The authors showed that L511Q is a functional hENaC variant that enhances Po [123]. In the hENaC structure [38], the site of this mutation lies in a bulge between β-strands 11–12 of the palm domain in the γ-subunit.

### 6.3. Gain-of-Function: Bronchiectasis

Bronchiectasis is a heterogeneous condition characterized by pathological dilation of the airways [139]. Bronchiectasis is one of the hallmark features of CF, where impaired mucociliary clearance in the airways is brought about by altered Cl(−) and Na(+) transport. Bronchiectasis with or without elevated sweat chloride-1 (BESC1 OMIM:211400) is caused by heterozygous mutation in the gene encoding the β-subunit of ENaC (*SCNN1B*). BESC1 is characterized by permanent airway dilation due to chronic bronchial inflammation or infection, and in some cases accompanied by abnormal sweat chloride concentration [140,141]. Bronchiectasis with or without elevated sweat chloride-2 (BESC2, OMIM:613021) is caused by mutation in the gene encoding the α-subunit of ENaC (*SCNN1A*), and bronchiectasis with or without elevated sweat chloride-3 (BESC3, OMIM:613071) is caused by mutation in the gene encoding the γ-subunit (*SCNN1G*). A number of patients with clinical symptoms of CF, but without mutations in the coding regions of the *CFTR* gene, have been investigated by different research groups [140,141,142]. In one early study of 20 non-classic CF patients without CFTR mutations and no renal symptoms, sequencing of the exons and flanking introns of genes encoding the α-, β- and γ-ENaC subunits identified six novel sequence changes [140]. Five of the six sequence changes caused amino acid changes, one in α-ENaC and four in β-ENaC; the sixth sequence change was at a splice junction in the *SCNN1B* gene. Functional characterization of three of the β-ENaC missense mutations in a *Xenopus* oocyte expression system revealed that two mutations generated lower Na(+) currents and one mutation higher Na(+) currents than wild-type αβγ-ENaC. Fajac et al. [141] screened the *SCNN1B* and *SCNN1G* genes in 55 patients with diffuse, idiopathic bronchiectasis and with only one or no *CFTR* mutations. In eight patients these researchers identified five different amino acid changes in β-ENaC and two in γ-ENaC. Three patients (of the 55) were transheterozygotes for CFTR and ENaC mutations, the p.Ser82Cys mutation in β-ENaC being present in all three patients. Subsequent screening of the *SCNN1A* gene in these patients revealed that one of the transheterozygote patients also carried the p.Trp493Arg gain-of-function mutation in α-ENaC. In the same study, the p.Trp493Arg gain-of-function mutation in α-ENaC was found in a female patient with idiopathic bronchiectasis and no mutation in the *CFTR* gene. In 55 patients with CF-like symptoms, but without any *CFTR* mutation, Mutesa et al. [143] identified five of eight ENaC variants previously found in five patients with a single CFTR mutation. Of these five ENaC variants, two occurred in *SCNN1A,* two in *SCNN1B* and one in *SCNN1G.*

Thus, there exists ample evidence that ENaC/CFTR genotypes contribute to lung disease and specifically bronchiectasis, and that a CF-like syndrome can be associated with CFTR and ENaC mutations acting either in concert or individually.

### 6.4. Loss-of-Function Mutations in ENaC: PHA-1B

At the other end of the spectrum from LS, CF and BESC is pseudohypoaldosteronism type-1 (PHA-1), a rare disease of mineralocorticoid resistance. Two forms of PHA-1 have been identified: the first is an autosomal dominant form with usually mild symptoms restricted to the kidney that is associated with heterozygous mutations in the *NR3C2* gene encoding the mineralocorticoid receptor [135,144]. The second type of PHA-1 is the generalized, systemic PHA-1 form, also called autosomal recessive PHA-1 (or PHA-1B), caused by loss-of-function mutations in the *SCNN1A*, *SCNN1B*, and *SCNN1G*, genes which encode α-, β- and γ-ENaC subunits, respectively [135].

PHA-1B (OMIM:264350) is a multisystem disorder characterized by salt wasting from the kidney, the colon, the sweat glands, and a reduced capacity to reabsorb Na(+) in the airways, leading to rhinorrhea, pulmonary congestion and recurrent pulmonary infections. In some cases, a skin rash (miliaria rubra) caused excessive salt in the sweat glands, is seen [145] (Figure 4).

PHA-1B usually manifests itself in the neonatal period and can be fatal in cases where severe electrolyte imbalance leads to cardiac failure or other complications [146]. Because of the multi-organ phenotypic consequences of PHA-1B and varying degrees of severity of symptoms, diagnosis is challenging [147,148,149,150]. Misdiagnosis is common, particularly as congenital adrenal hyperplasia (CAH) [151] or CF [148,152]. Although PHA-1B can be life-threatening, survival with normal physical and neurological outcome is possible when correct management prevents electrolyte imbalance [147]. A correlation between severity of phenotype and genotype has not been possible so far for PHA-1B owing to the small number of cases reported, the rarity of the disease and the heterogeneity of symptoms. Communication of genotypic and phenotypic data from clinical cases through the variant databases is of paramount importance to help clinicians worldwide not only to diagnose and treat this disease, but also to establish whether a genotype–phenotype correlation is discernible.

According to reports in the literature, about 70 unique mutations have been found to cause PHA-1B in individual patients who are homozygous or compound heterozygous for these variant ENaC genes (see Table 1 and Appendix A). The most comprehensive register of ENaC variants which have been reported in the literature to cause PHA-1B in individual patients can be found in the Global Variome shared LOVD database [153].

It may be confidently assumed that there are more variants of ENaC observed to cause PHA-1B in patients, but they have neither been reported in scientific publications nor registered in the variant databases. PHA-1B is classified as an orphan disease owing to its rare occurrence, and it is most often, but not always, observed in communities or populations where parents are consanguineous, with the recessive mutant allele occurring in both asymptomatic, heterozygous parents.

Amongst the 70 or so ENaC variants known to cause PHA-1B, 41 occur in *SCNN1A*, encoding the α-subunit, 21 in *SCNN1B*, encoding the β-subunit and 7 in *SCNN1G,* encoding the γ-subunit (Table 1). The markedly higher frequency of mutations observed in *SCNN1A* (41 of the 69 known variants) likely reflects the crucial importance of the α-subunit of ENaC in forming a functional ion channel [39,47,154]. Furthermore, the relative proportion of frameshift mutations occurring in *SCNN1B* (10 out of 21) is higher than that for *SCNN1A* (14 out of 41) and *SCNN1G* (2 out of 7), likely reflecting the different functions of each subunit in the ENaC heterotrimer (Table 1).

The distribution of the mutations in the 3D structure of ENaC subunits is not random, but rather mutations are found in locations of the architecture which have some functional importance or significance (See Appendix A).

## 7. ENaC and Disease Conditions

Deletion of genes encoding subunits of ENaC results in early postnatal mortality. The critical role of ENaC in maintenance of blood pressure, electrolyte and fluid balance becomes obvious by the occurrence of disease conditions in which ENaC function is disrupted by mutation. Mutations in ENaC that cause hereditary diseases such as LS, PHA-1B, bronchiectasis and CF present in different phenotypes.

### 7.1. Role of ENaC in Salt-Sensitive and Resistant Hypertension

Resistant HTN is an important and preventable cause of stroke as well as cardiovascular and kidney disease. Uncontrolled HTN is not uncommonly due to non-compliance, but also consumption of substances that elevate blood pressure such as high-salt diet and finally therapeutic and diagnostic failure may contribute.

Several genes have been identified as playing a role in the genetics of HTN such as Sodium Channel Epithelial 1 subunits (*SCNN1A*, *SCNN1B*, *SCNN1C*) and recently reported *SCNN1D* [155], Armadillo Repeat Containing 5 (*ARMC5*), G Protein-Coupled Receptor Kinase 4 (*GRK4*) and Calcium Voltage-Gated Channel Subunit A1 D (*CACNA1D*) [156]. Knowledge of genetic differences in HTN is crucial for effective personalized pharmacotherapy.

Upon increased sodium intake some individuals react with a significant rise in blood pressure while others experience almost no change. At least in part, these differences are due to genetic variation of ENaC. In the Genetic Epidemiology Network of Salt-Sensitivity (GenSalt) Study a relationship between rare variants in the ENaC pathway and blood pressure salt-sensitivity was demonstrated [157]. In a further GenSalt Study, 1906 participants had a one week low-sodium diet, followed by a high sodium diet for another seven days. Then the most salt-sensitive and most salt-resistant study participants were selected for resequencing of the three ENaC genes *SCNN1A*, *SCNN1B*, and *SCNN1G*. Neither *SCNN1B* nor *SCNN1G* were found to be associated with salt-sensitive HTN, while analyses of *SCNN1A* revealed three independent common variants, rs11614164, rs4764586, and rs3741914, which were associated with salt-sensitivity [158]. Since then, a growing list of benign and pathogenic mutations can be found in the ClinVar database; some of the pathogenic mutations cause LS which manifests as a severe form of HTN. Recently, in seven out of thirteen probands from a Czech family a novel mutation in the β-subunit of ENaC encoded by the *SCNN1B* gene was identified. The nonsense mutation carriers in the protein sequence p.Tyr604* differed in the severity of HTN and hypokalaemia, whereas hypoaldosteronemia was a sensitive sign in all mutation carriers [159]. It is well established that mutations causing LS result in increased activity of ENaC. Most of them cause disruption or loss of the PY motif of β- or γ-subunits and therefore impaired channel degradation [160]. Mutations causing disruption of a disulfide bridge in the extracellular loop of the α-subunit, leading to elevation of channel Po, has also been described as resulting in LS [84,160] (described previously).

Mutchler et al. [55] recently reviewed the role of ENaC in the development of salt-sensitive HTN, presenting the function of ENaC in brain, vasculature and immune cells. In the brain, ENaC is particularly located in vasopressin magnocellular neurons in the hypothalamic supraoptic and paraventricular nuclei, where α-, β- and γ-ENaC subunits were detected. In these tissues ENaC mediates a sodium leak current that affects the steady state membrane potential in vasopressin neurons. As vasopressin neurons have a crucial role in the coordination of neuroendocrine and autonomic responses to maintain cardiovascular homeostasis, the effect of dietary salt intake on ENaC regulation and activity in vasopressin neurons was further studied by Sharma et al. [26]. High dietary salt intake induced an enhanced expression of β- and γ-ENaC subunits in the supraoptic nucleus as well as translocation of α-ENaC towards the plasma membrane. Compared to control recordings, the mean amplitude of the recognized ENaC currents was significantly greater in vasopressin neurons from animals that were fed a high salt diet. Partly due to the increased ENaC current the basal membrane potential in vasopressin neurons became more depolarized in the high salt diet group. These findings were interpreted such that high dietary NaCl intake enhances the expression and activity of ENaC, followed by an augmentation of the synaptic drive depolarizing the basal membrane potential close to the action potential threshold. Noteworthy however, kinetic analysis of the ENaC current reveals only a minor role in the regulation of the firing activity of vasopressin neurons in the absence of synaptic inputs. These findings support the assumption that ENaC in the brain plays an important role in blood pressure regulation [26].

The enzyme Cyp2c44 plays a role in salt-sensitive HTN as it was shown that the lack of this enzyme caused HTN in *Cyp2c44*(-/-) mice on a high salt diet. In these mice, increased gating of the ENaC current was observed in the dissected kidney collecting duct principal cells. The ENaC inhibitor amiloride and the Cyp2c44 epoxygenase metabolite 11,12-epoxyeicosatrienoic acid were able to lower the blood pressure of hypertensive *Cyp2c44*(-/-) mice [161]. Alterations in the expression and/or activities of CYP2C8 and/or CYP2C9, the human functional homologues of Cyp2c44, could play a role in the pathophysiology of human HTN. This must be kept in mind when targeting Cyp2c44 epoxygenase, because several drugs in current clinical use are metabolized by those human CYP isoforms.

An Ang II- and aldosterone-induced increase in the anion exchanger protein pendrin contributes to blood pressure regulation through Cl(−) absorption and an indirect effect on ENaC [162]. In mouse models of LS, pendrin gene ablation did not change ENaC subunit total protein abundance, subcellular distribution, or channel density, but markedly reduced channel Po [163]. To study whether K(+) loss is reduced by ENaC downregulation, pendrin knockout (KO) mice received a Na(+), K(+), and Cl(−)-deficient diet. In this mouse model, hypokalemia was aggravated with ENaC stimulation and abolished with ENaC inhibition. However, the prevention of K(+) loss by reduced ENaC activity counteracts the stimulation of ENaC required for appropriate regulation of blood pressure and intravascular volume. Notably, in pendrin KO mice NaCl restriction caused less ENaC stimulation than in wild-type mice [164].

#### 7.1.1. Role of ENaC in HTN with Comorbidities

Diabetic nephropathy is a common complication in patients suffering from diabetes mellitus, and is associated with HTN, proteinuria and excretion of urinary plasmin. Andersen et al. [165] postulate plasmin-induced promotion of HTN with albuminuria, probably through ENaC. Thus, these researchers used diabetic, male plasminogen-deficient and wild-type Ang II-treated mice in their study. In diabetic wild-type mice, plasma and urine glucose concentrations as well as urine albumin and plasminogen excretion were increased, while diabetic plasminogen-deficient mice showed albuminuria, but no plasminogen in the urine. Ang II elevated blood pressure in wild-type, diabetic wild-type and plasminogen-deficient control mice, whereas Ang II did not change blood pressure in diabetic plasminogen-deficient mice. In wild-type Ang II-treated diabetic mice blood pressure could be reduced upon amiloride administration. Independent of diabetic condition, the urine from wild-type mice generated larger amiloride-sensitive current than urine from plasminogen-deficient mice. While full-length γ-ENaC and α-ENaC subunit abundances were not changed in kidney homogenates, γ-ENaC cleavage product was increased in diabetic compared with non-diabetic mice, demonstrating the involvement of plasmin in ENaC activation [165].

In kidney transplant recipients, albuminuria associated with HTN is a predictor for adverse renal outcome. In a cross-sectional study increased urinary serine proteases were detected in these patients enabling ENaC activation with increased abundance of γ-ENaC [166]. Ray et al. [167] characterized a mouse with significantly suppressed expression of the γ-ENaC in order to prove the importance of this subunit in homeostasis of electrolytes and body fluid volume. To study this task, they used the hypomorphic (*γ*^mt^) allele of *γ*^mt/mt^ mice. In control mice, high-salt diet caused a transient increase in body water, while in *γ*^mt/mt^ mice lower blood pressures were variably detected. In *γ*^mt/mt^ mice on a high Na(+) diet, non-dipping of the blood pressure was observed. As a result of the observations with their mouse model, the authors suggest that ENaC in tissues other than the kidney may participate in the regulation of extracellular fluid volume and blood pressure beyond classical transepithelial Na(+) transport mechanisms [167].

In chronic kidney disease patients, NADPH oxidase (NOX)-derived ROS are frequently involved in treatment-resistant HTN. NOX activator 1 (NOXA1) is located in epithelial cells of the loop of Henle’s thick ascending limb and distal nephron. Ang II increased NOXA1/NOX1 expression and induced ROS in the kidney of male wild-type mice by activating this enzyme, which further caused enhanced tubular ENaC expression and Na(+) reabsorption, generating an increase in blood pressure. Genetic deletion of *NOXA1* subunit of NOX1 reduced both basal and Ang II-induced HTN. The attenuation of Ang II-induced HTN observed in female mice is thought to be due to weaker NOXA1/NOX1-dependent ROS signalling and efficient sodium excretion. In a mouse renal epithelial cell line, aldosterone induced ROS, *NOXA1* and *SCNN1A* expression as well as ENaC activity, which could be blocked by *NOXA1* small-interfering RNA [120].

Decreased urinary Na(+) excretion was observed in hyperuricemic rats, which was antagonized by amiloride application. Elevated serum uric acid caused high blood pressure and renal tubulointerstitial injury. In addition, in hyperuricemic rats the increased expressions of α-, β- and γ-ENaC subunits, SGK1, and glucocorticoid-inducible leucine zipper protein 1 (GILZ1) were found. As a consequence of its uricosuric activity, benzbromarone could prevent these effects. Therefore, the authors suggest that elevated serum uric acid induces HTN by ENaC activation and affects the ENaC regulatory complex components [168].

Quadri et al. [169] postulate that obesity induced Na(+) retention and HTN are mediated Ang II-independently via the renal (pro)renin receptor-SGK-1-α-ENaC pathway. In mice, a high-fat diet caused significant increases in systolic blood pressure and body weight, and significant reductions in urine volume and urine Na(+) concentration. The authors showed that, compared to a regular diet with 12 kcal% fat, a high-fat diet with 45 kcal% fat significantly increased mRNA and protein expression of renal (pro)renin receptor, α-ENaC, p-SGK-1 and Ang II. Whereas in nephron specific renal (pro)renin receptor knockout mice, reduced mRNA and protein expression of (pro)renin receptor, p-SGK-1 and α-ENaC was observed, accompanied by an increased urine volume and urine Na(+) as well as significantly reduced systolic blood pressure [169].

Besides being implicated in HTN, cardiovascular and kidney disease excessive salt intake can also promote bone resorption. In one study, high salt diet in ovariectomized Sprague Dawley rats was applied as a construct for a high bone turnover model. Expression of α-ENaC and voltage-gated chloride channels was up-regulated, whereas expression of sodium-chloride co-transporter and sodium-calcium exchanger was down-regulated in femoral tissue and renal tubules indicating that high salt diet can not only cause HTN, but can also destroy the microstructure of bone by increasing bone resorption, affecting some ion channels of bone tissue [31].

Preeclampsia is characterized by HTN, proteinuria, suppression of RAAS, and impaired urine sodium excretion. Plasma and urine was sampled from patients with preeclampsia, healthy pregnant controls and non-pregnant women [170]. Findings revealed that pregnancy and preeclampsia were associated with increased abundance of furin-cleaved α-ENaC subunit causing Na(+) reabsorption and urine K(+) loss [170].

#### 7.1.2. ENaC Expression in Blood Cells of Hypertensive Patients

Abnormalities in platelet functions, such as platelet hyperactivity and hyperaggregability, contribute to thrombotic complications in hypertensive patients. Platelets are small, anucleated cell fragments that become activated in response to a wide variety of stimuli, initiating intracellular pathways which cause haemostatic thrombus formation at vascular injury sites. In essential HTN, platelet activation contributes to the development of myocardial infarction and ischemic stroke. Biochemical, cell and molecular biology results demonstrated that ENaC is overexpressed in the platelets from hypertensive patients, and that structural and biochemical abnormalities in lipid membrane composition and fluidity characteristic of platelets influence the expression of ENaC and differ from healthy individuals. Additionally, results strongly suggest a key role of β-dystroglycan as a scaffold for the organization of ENaC and associated proteins such as caveolin-1 [28].

As overexpression of ENaC on the plasma membrane of human platelets is a characteristic of arterial HTN, a double-blinded study was initiated in order to investigate the sensitivity and specificity of a diagnostic assay for expression of ENaC in platelets. For this purpose, gold nanoparticles were conjugated to an antibody against ENaC. In 59.7% of patients with an undiagnosed HTN, an increased blood pressure could be detected, so that an application of a refined version of this assay for initial screening and early diagnosis of HTN is recommended [171].

#### 7.1.3. Effect of ENaC on Vascular Response in Hypertensive Conditions

ENaC in endothelial tissue has a role in regulating vascular tone [23]. High salt- and fat-intake associated with aldosterone cause vascular stiffening in humans. Studies in female mice on diet with high saturated fat and high refined carbohydrates revealed that enhanced endothelium mineralocorticoid receptor signalling induced expression and translocation of α-ENaC to the endothelial cell surface, generating vascular fibrosis and stiffness which could be antagonised by amiloride. This finding was validated by using ENaC α-subunit knock-out mice and supports the assumption that α-ENaC subunit activation contributes to observed vascular responses [172].

Nakamura et al. [173] propose the intestinal mineralocorticoid receptor as a target for studying the molecular mechanism of blood pressure regulation and cardiovascular diseases, because this receptor regulates intestinal Na(+) absorption in the colon and is linked with ENaC expression. To prove the role of α-ENaC and exogenous mineralocorticoid, Zhang et al. [22] used either mice with specific deletion of the α-ENaC subunit or treated them with an mTOR- inhibitor. The protein mTOR is a downstream signalling molecule that is involved in mineralocorticoid receptor activation of ENaC. With the high-salt diet, DOCA (deoxycorticosterone acetate)-treated control mice developed increased blood pressure and arterial stiffness, and enhanced sodium transport activity, while treatment with an mTOR inhibitor or deletion of α-ENaC attenuated these effects and prevented salt-induced impairment of vascular relaxation, supporting the role of α-ENaC and mineralcorticoids in the development of vascular stiffening with a high-salt diet.

Arterial stiffness and impaired vasorelaxation contribute to deterioration of insulin resistance and finally diabetes. Besides alterations in extracellular vesicles and their microRNAs, abnormal gut microbiota and increased renal sodium-glucose cotransporter activity, development of HTN in diabetic patients is also promoted by enhanced activation of ENaC [174]. As elevated expression and increased activity of vascular ENaC can result in vascular dysfunction which had already been proven in small animal models, the expression and function of ENaC in human vasculature was studied by Paudel et al. [21] in the human internal mammary artery and aorta obtained from patients undergoing coronary artery bypass graft surgery. In these preparations, expression of α-, β-, γ- and δ-subunits was detected at mRNA and protein levels. Single channel conductance suggested the existence of the αβγ- and δβγ-ENaC configuration in the vasculature. Subunit expression levels were compared between arteries from normotensive, uncontrolled hypertensive and controlled hypertensive patients. In controlled hypertensive patients, reduced expression of the δ-ENaC subunit was observed in the internal mammary artery, while in the aorta, reduced expression of γ-ENaC was found which implies an association of ENaC expression levels with HTN [21]. Although *SCNN1D* is poorly expressed in human kidney tissue, *SCNN1D* variants were associated with systolic blood pressure, diastolic blood pressure, mean arterial pressure, and pulse pressure, suggesting that variants in extrarenal ENaCs, in addition to ENaCs expressed in kidneys, influence blood pressure and kidney function [155].

Moreover, high salt intake stimulates endothelial cells to express and release bone morphogenetic protein-4 (BMP4), leading to reduced vascular relaxation by stimulating ENaC in a benzamil-sensitive manner. Stimulation is mediated by p38 mitogen-activated protein kinases (p38 MAPK) and SGK1/NEDD4-2. These data suggest stimulation of ENaC in endothelial cells by BMP4 as an additional pathway to participate in the complex mechanism of salt-sensitive HTN [175].

Because DEG/ENaC proteins such as β-ENaC are evolutionarily linked to known mechanosensors, the expression of β-ENaC has been studied in the pressure-induced constriction of arteries and arterioles, and indeed it could be shown that up-regulated β-ENaC enhances pressure-induced constriction in isolated middle cerebral artery. Thus, it is hypothesised that inflammation-mediated down-regulation of β-ENaC might contribute to cerebrovascular dysfunction which has to be further studied. In addition, the findings show that expression of exogenous β-ENaC increases pressure-induced constriction in middle cerebral artery without affecting overall contractility. This effect could be used to study the impact of specific proteins which are down-regulated in disease models where pressure-induced vasoconstriction responsiveness is deteriorated, such as β-ENaC in preeclampsia [14].

Noteworthy, studies on different types of resistance arteries provide evidence for different roles ofENaC, and the reasons for these discrepancies still must be elucidated [176]. While α-ENaC knockout mice lacked an aldosterone-dependent reduction in acetylcholine-induced vasodilation and were resistant to vascular stiffening as confirmed by pharmacological studies, there is some inconsistency concerning the effects of pharmacological versus genetic inhibition of ENaC on acetylcholine-induced NO production [55,168].

#### 7.1.4. Shear Force Sensing of ENaC

Sensing of mechanical force in endothelial cells is essential for a normal vascular function. Po of ENaC is directly regulated by laminar shear stress [177] implying that local changes of shear stress can directly influence Na(+) influx through the channel. Apart from ENaC, the Transient Receptor Potential (TRP) channel, Acid Sensing Ion Channels (ASIC) and Piezo channels [178] act as mechanosensors. These ion-transporting proteins are involved in baroreflex control and have been found in the walls of the aortic arch and carotid sinuses, as well as in brain astrocytes. These mechanosensors are essential for the rapid moment-to-moment feedback regulation of blood pressure [179]. There is strong evidence that ENaC-mediated shear force responsiveness depends on the “force-from-filament” principle involving extracellular tethers and matrix. In particular, two glycosylated asparagines localized in the palm and knuckle domains of α-ENaC are important for shear force sensing [50]. Although it is established that N-glycans attach to the glycosylated asparagines of α-ENaC, the mode in which the N-glycans mediate this interaction requires further investigation. ENaC as an arterial shear stress sensor is connected to an intact endothelial glycocalyx which enables NO production [180]. The signalling pathways downstream of this event are still not completely understood as interaction and cross talk between these pathways is a complex phenomenon. There is evidence that eNOS is one potential contributor (e.g., [176,181,182]), but on the contrary Ydegaard et al. [183] found that the antihypertensive effect of amiloride occurs independently of ENaC and eNOS in human femoral arteries and veins in mice.

#### 7.1.5. Effect of ENaC on Inflammatory Responses in Hypertensive Conditions

ENaC is expressed in dendritic cells and contributes to immune system activation [23]. In a recent review article, Ertuglu and Kirabo [56] thoroughly describe the potential role of increased systemic inflammatory biomarkers such as C-reactive protein, interleukin (IL)-6 and tumor necrosis factor (TNF)-α, as well as inflammatory infiltration of the renal interstitium and vascular wall in the development of HTN and organ damage. Briefly, high concentrations of extracellular sodium can directly trigger an inflammatory response in antigen-presenting cells. Sodium influx into antigen-presenting cells is mediated by ENaC and implies an immune-dependent modulatory effect of ENaC on blood pressure in extra-renal tissue [56,184].

Chronic renal inflammation has been proven to promote high blood pressure in humans and animals. Veiras et al. [185] identified IL-1β from renal tubular epithelial cells, but not from immune cells, as an initial promoter of renal inflammation, and hence they proposed this finding as a novel molecular mechanism regarding the development of renal inflammation and salt sensitivity during diabetes. They showed that specific suppression of IL-1β in renal tubules prevented salt sensitivity in diabetic mice, which exhibit significantly higher levels of IL-1β in renal tubules than non-diabetic mice. Accordingly, a primary culture of renal tubular epithelial cells from wild-type mice released significant levels of IL-1β upon exposure to high glucose. Co-culture of tubular epithelial cells and bone marrow-derived macrophages revealed that tubular epithelial cell-derived IL-1β promotes the polarization of macrophages towards a proinflammatory phenotype, resulting in IL-6 secretion. Consequently, these findings were also evaluated in vivo in diabetic mice which were transplanted with the bone marrow of IL-1 receptor type 1 knockout mice. These mice were salt-resistant, displayed lower renal inflammation and lower ENaC expression and activity compared with wild-type bone marrow transplanted diabetic mice [185].

Studies on salt sensitivity of blood pressure revealed that besides renal epithelium and aldosterone-mediated activation of ENaC, myeloid immune cells also recognize sodium ions via ENaC, initiating activation of fatty acid oxidation and the nicotinamide adenine dinucleotide phosphate oxidase enzyme complex, which produced superoxide with the subsequent formation of immunogenic isolevuglandin-protein adducts [186,187]. Entry of sodium ions into dendritic cells through α- and γ-ENaC and the sodium hydrogen exchanger 1 causes calcium influx via the sodium-calcium exchanger, activation of phosphokinase C, phosphorylation of p47*^phox^* and association of p47*^phox^* with gp91*^phox^*. When dendritic cells are activated by high sodium they increase IL-1β production and promote T-cell production of cytokines IL-17A and interferon-γ. These findings present a mechanistic link between salt, inflammation, and HTN [186].

ENaC has a key role in inflammatory response as it is also crucial for actin-cytoskeleton reorganization during polarization and directed migration, although expression of sodium channels other than ENaC in the neutrophil activated state cannot be excluded. Overexpression of ENaC promotes sodium influx and an increase in calcium, leading to amplification of the active state of neutrophils, which initiates oxidative stress and damage to the endothelium. It is granted that neutrophils from hypertensive patients defend against pathogen infections, but on the other hand they contribute to chronic inflammation in HTN [29].

#### 7.1.6. Putative Targets and ENaC-Modulating Compounds for Treatment of Cardiovascular Disease

New data derived from studies about structure, regulation and expression of ENaC not only provide knowledge about the complex mechanisms of HTN development, but also reveal potential new drug targets for treatment of cardiovascular diseases:Amiloride binding site (ENaC inhibitor amiloride and derivatives)Extra-renal ENaC including vascular and immune cellsDe-ubiquitination enzyme activating agentsIntestinal mineralocorticoid receptorInwardly rectifying K(+) (K_ir_) channelsInhibition of ROS activation (NOX subunit-specific inhibitors, tempol, quercetin, allicin)Synthetic serine protease inhibitors (camostat)

##### Amiloride Binding Site—ENaC Inhibitors

Individualised therapy of HTN based on phenotyping by plasma renin activity and aldosterone as well as overactivity of renal ENaC can markedly improve blood pressure control. Spence [188] assumes that ENaC overactivity is far more common than most physicians suppose. However, administration of ENaC inhibitors such as amiloride is not a routinely used treatment option because of its low efficacy when compared with other diuretics. However, a metaanalysis by Hebert et al. [189] shows that treatment of elderly hypertensive patients with ENaC inhibitors combined with a thiazide diuretic reduced coronary mortality and sudden cardiac death. Furthermore, a Nigerian study was able to demonstrate that amiloride is an effective antihypertensive in patients carrying polymorphisms of the β-subunit of ENaC [190].

To determine whether treatment with amiloride improves endothelial function and arterial stiffness in obese insulin resistant subjects, a randomized placebo-controlled trial (NCT03837626) examining pre- and postmenopausal women and age-matched men is currently recruiting (Table 2). So far, amiloride is mainly applied as an add-on drug in combination with established antihypertensives. Table 2 lists clinical trials conducted with amiloride. Most of these investigate the effect on resistant HTN, some in combination with comorbidities. However, as yet no data are available from these studies.

Antiangiogenic receptor tyrosine kinase inhibitors such as sunitinib are known to increase blood pressure. In order to elucidate the underlying mechanism of this unwanted side effect Witte et al. [191] studied the effect of sunitinib on sodium reabsorption, renal ENaC and sodium chloride cotransporter in rats. After two weeks of treatment with sunitinib only an increase in renal medullary β-ENaC protein abundance was observed, while α- and γ-ENaC as well as the sodium chloride cotransporter did not differ from control rats. The increase in blood pressure could be reduced by amiloride or hydrochlorothiazide, but a combined application of these two drugs did not improve the outcome. Thus, the authors conclude that ENaC-dependent and thiazide-sensitive sodium-retaining mechanisms are not responsible for the sunitinib-induced HTN, but that in this setting ENaC blockers as well as thiazides are useful drugs [191].

##### Extra-Renal ENaC including Vascular and Immune Cells

Hitherto most of the studies on ENaC have focused on renal Na(+) and K(+) regulation. But the recent discoveries of the existence and functioning of extra-renal ENaC including vascular smooth muscle and immune cells may shed light on novel therapeutic targets for ENaC in salt-induced cardiovascular disease [53].

Noteworthy, δ-ENaC is expressed in various mammalian species, except mice and rats, although these are common animal models for cardiovascular research. However, the recent discovery of δ-subunit in human vascular cells indicates that this subunit may play a significant role in normal and pathological vascular physiology in humans. Channels containing the δ-subunit have different biophysical and pharmacological properties compared with channels containing the α-subunit, with the potential to alter the vascular function of ENaC in health and disease. Thus, it is important to investigate the expression and function of δ-ENaC in the human vasculature more thoroughly to identify whether δ-ENaC can serve as a potential new drug target [20].

Pitzer et al. [53] reviewed recent studies on how ENaC is regulated in both the kidney and other sites including the vascular smooth muscles, endothelial cells, and immune cells.

##### De-Ubiquitination Enzyme Activating Agents

The review article of Ishigami et al. [52] describes the involvement of the E3-ubiquitin ligase Nedd4-2/NEDD4L in salt-sensitive HTN spanning from detailed genetic dissection analysis to the development of a genetically engineered mouse model. The ENaC-NEDD4L system plays an important role in post-translational modification through ubiquitination, which regulates ENaC expression on the cell membrane of the terminal nephron. Thus, the researchers [52] suggest human NEDD4L as another possible candidate gene for salt-sensitive HTN, and that gain of knowledge about the NEDD4L gene will enable therapeutic and diagnostic applications, making de-ubiquitination enzyme activating agents for antagonism of human NEDD4L a new interesting drug target. However, ubiquitination is an extremely complex, temporally controlled and highly regulated process that can play major roles in various pathways during health and disease conditions which impedes specific targeting.

##### Intestinal Mineralocorticoid Receptor

The intestinal mineralocorticoid receptor regulates intestinal sodium absorption in the colon and contributes to blood pressure regulation. For their study Nakamura et al. [173] used intestinal epithelial cell-specific mineralocorticoid receptor knockout mice which had markedly decreased colonic expression of β- and γ-ENaC. The increase in salt-sensitive blood pressure was significantly smaller in knockout mice than in control mice. These findings suggest the intestinal mineralocorticoid receptor as a possible target for studying the molecular mechanism of HTN and cardiovascular diseases, and probably for their treatment.

##### Inwardly Rectifying K(+) (K_ir_) Channels

Inwardly rectifying K(+) (K_ir_) channels located on the basolateral membrane of epithelial cells of the distal nephron play a crucial role in K(+) handling and blood pressure control, making these channels an attractive target for the treatment of HTN. Isaeva et al. [192] determined how the inhibition of the basolateral K_ir_ 4.1/K_ir_ 5.1 heteromeric K(+) channel affects ENaC-mediated Na(+) transport in the principal cells of the cortical collecting duct. They found that the inhibition of K_ir_ 4.1/K_ir_ 5.1, but not of the K_ir_ 4.1 channel, depolarizes the cell membrane. Thereby elevation of the intracellular Ca(2+) concentration is induced and ENaC activity is suppressed. These data present a specific role of the K_ir_ 4.1/K_ir_ 5.1 channel in the modulation of ENaC activity and underline the potential of K_ir_ 4.1/K_ir_ 5.1 inhibitors to regulate electrolyte homeostasis and blood pressure via modulation of ENaC [192].

##### Inhibition of ROS Activation (NOX Subunit-Specific Inhibitors, Tempol)

Salt-sensitive HTN is associated with an increase in the production of ROS, which are known to increase the activity of ENaC, and therefore indirectly affect Na(+)-retention and increase blood pressure. Chronic disease-related activation of renal NOXA1/NOX1 possibly causes enhanced Na(+) reabsorption and increased blood pressure without apparent RAAS hyperactivation, making widely used drugs like angiotensin-converting enzyme inhibitors, angiotensin receptor blockers or mineralocorticoid receptor antagonists ineffective. Thus, therapeutic interventions that reduce NOXA1/NOX1 activation and ROS generation in renal epithelial cells and thereby inhibit ENaC hyperactivity may be advantageous in the treatment of chronic kidney disease, diabetes, and therapy resistant HTN. Setanaxib as a member of the novel drug class of naxibs is a specific dual NOX1 and NOX4 inhibitor, preventing the excessive activation of MAPK pathway induced by Nox1/4-derived ROS, and further leading to the delay or prevention of progression of many cardiovascular disorders including the improvement of doxorubicin-induced cardiotoxicity [193]. Although NOXA1/NOX1 is a promising target in drug discovery, so far, no NOX subunit-specific ENaC inhibitors are currently available for therapeutic use. However, the newly developed drug GKT137831, aka setanaxib (Table 3), has already produced encouraging results in cancer [194,195], primary biliary cholangitis [196] and apparently also HTN [119,193].

The potent antioxidant tempol (4-Hydroxy-2,2,6,6-tetramethylpiperidine-N-oxyl) is a superoxide dismutase mimetic which has been shown to decrease blood pressure in various animal models of HTN. This effect was accompanied by vasodilation, increased nitric oxide activity, reduced sympathetic nervous system activity at central and peripheral sites and enhanced potassium channel conductance in blood vessels and neurons [197]. In salt-sensitive hypertensive 129Sv mice, administration of the antioxidant tempol caused reduced systolic blood pressure as well as a decrease in renal protein expression of the α-ENaC subunit and its adaptor protein MLP-1 [198].

Even the natural antioxidant food ingredients quercetin, ginger and allicin were shown to have an effect on ENaC. The flavonoid quercetin activates the sodium-potassium-chloride co-transporter 1 in renal epithelial cells and consequently elevates the chloride concentration. This causes reduced expression of ENaC, leading to a decrease in renal Na(+) reabsorption which in turn results in lowering of elevated blood pressure [199]. Garlic (*Allium sativum*) was found to decrease transmembrane currents of ENaC-expressing oocytes in a dose-dependent and irreversible manner. The effect of garlic was blocked by dithiothreitol and L-cysteine which suggests an involvement of thiol-reactive compounds, and indeed the thiol-reactive garlic compound allicin significantly inhibited ENaC. Conversely, the garlic organosulfur compounds S-allylcysteine, alliin and diallyl sulfides had no effect on ENaC [200]. The mechanism of the antihypertensive effect of 6-gingerol, one of the main ingredients of ginger, is suggested to be caused by an increased levels of phosphorylated endothelial nitric oxide synthase (eNOS) protein in vascular endothelial cells and down-regulated ENaC in kidney cells through PPAR𝛿 regulation [201]. However, findings in human femoral arteries and veins in mice question the role of ENaC and eNOS in the context of antihypertensive action [183].

Despite these promising results in pre-clinical studies, in clinical trials, the elimination of ROS by the use of unspecific antioxidant compounds was not successful in the prevention or therapy of diseases. Consequently, this demands controlling specific ROS-mediated signalling pathways by selective targeting such as NOXA1. This could offer a perspective for a more refined redox medicine including enzymatic defence systems [202].

##### Synthetic Serine Protease Inhibitors

The endogenous prostasin inhibitor nexin-1 inhibits ENaC activity through the suppression of prostasin activity. The same applies to the synthetic serine protease inhibitor camostat mesilate which decreases the ENaC current as a result of reduced prostasin activity. When orally administered to Dahl salt-sensitive rats, camostat mesilate caused a significant decrease in blood pressure accompanied by an elevated urinary sodium/potassium ratio. Based on these data, synthetic serine protease inhibitors might represent a new class of antihypertensive drugs in patients with salt-sensitive HTN [203].

### 7.2. Role of ENaC in Cystic Fibrosis

#### 7.2.1. Pathology and Therapeutic Approach

Cystic fibrosis, an autosomal recessive disorder caused by mutations in the cystic fibrosis transmembrane conductance regulator (CFTR) gene that codes for the CFTR chloride channel, is one of the most common life-shortening hereditary diseases affecting the lungs and other organs. In physiological conditions, airway surface liquid hydration is maintained by a fine-tuned balance between ENaC-mediated Na(+) absorption and CFTR-dependent chloride secretion. CFTR and ENaC are within reach of each other, suggesting a direct intermolecular interaction between these two proteins [204]. In airways of CF patients, the dysfunction of CFTR is accompanied by increased ENaC activity, leading to airway surface liquid dehydration which causes disturbed mucus viscosity and accumulation with inflammation which favours chronic bacterial infection and respiratory failure.

CFTR and ENaC also play an important role in the pathogenesis of chronic rhinosinusitis [205] and bronchiectasis [140,141]. Chronic rhinosinusitis is an inflammatory disease of the nose and the paranasal sinuses, often associated with an infection by *Staphylococcus aureus.* Increased mRNA expression level of CFTR and decreased level of ENaC can be modulated by *S. aureus* infection and budesonide treatment [205]. There is also evidence that bronchiectasis, which is characterized by permanent airway dilation due to chronic bronchial inflammation or infection, and in some cases accompanied by abnormal sweat chloride concentration, is caused by mutation in the genes encoding for ENaC. Based on their findings, Mutesa et al. [143] suggest that the CF-like syndrome could be associated with the concomitant occurrence of CFTR and ENaC mutations, however was also identified in patients without CFTR mutations.

ENaC has also a key role for inflammation in CF as it is crucial for actin-cytoskeleton reorganization during polarization and directed migration. Overexpression of ENaC promotes sodium influx and an increase in calcium leading to amplification of the active state of neutrophils, which initiates oxidative stress and damage to the endothelium [29]. Further, it is postulated that NLRP3 inflammasome activation and ENaC upregulation drives enhanced innate-immune responses as indicated by increased levels of IL-18, IL-1β, caspase-1 activity and release of apoptosis-associated speck-like (ASC) protein aggregates in monocytes from patients with CF-associated mutations. These findings were reversed by pre-treatment with inhibitors of NLRP3 inflammasome and ENaC inhibitors as well. Moreover, overexpression of β-ENaC also increased NLRP3-mediated inflammation in the absence of CFTR dysfunction, indicating a role for sodium in modulating NLRP3 inflammasome activation [206].

Elevated high-mobility group box-1 protein (HMGB-1) levels and hyperactivity of ENaC are hallmark features of the CF lung. HMGB-1 signalling to the receptor for advanced glycation end products plays an important role in the maintenance of ENaC dysfunction and inflammation in the CF lung [207].

Pathology, mechanisms underlying CF and novel therapeutic approaches have been extensively reviewed recently [54,208,209,210,211,212,213,214,215].

For the *CFTR* gene more than 2000 different CF-causing mutations have been identified. The impaired electrolyte homeostasis caused by the mutated or absent protein leads to symptoms in multiple organ systems with predominantly pulmonary manifestation. Formerly, only symptomatic treatment was available. However, in recent years, different combinations of CFTR modulators have become available for patients with common mutations, including the CFTR potentiator ivacaftor and combinations of correctors (lumacaftor, tezacaftor, elexacaftor). Treatment with these drugs improves life expectancy and quality of life of CF patients. However, some patients with rare *CFTR* mutations gain no benefit from treatment with the combination of CFTR modulating drugs [216,217].

Given the hyperactivity of ENaC in CF, ENaC inhibitory agents have been investigated as potential therapeutic agents.

Potential new drug targets for treatment of CF include:ENaC inhibitors (BI 1265162, GS-9411, AZD5634)SPLUNC1 mimetics (SPX-101)Epigenetic technologyENaC genes (antisense oligonucleotides, siRNAs)Proteolytic cleavage (CAPs inhibitor QUB-TL1)Inhibition of myeloperoxidase-derived oxidants (AZM198)

#### 7.2.2. Compounds Targeting ENaC (ENaC Inhibitors and SPLUNC1 Mimetics)

ENaC, the calcium-activated chloride channels Transmembrane Protein 16A (TMEM16A) and Solute carrier family 26 member 9 (SLC26A9) and the proton pump Potassium-transporting ATPase alpha chain 2 (ATP12A) have been associated with CF pathology. Thus, they are proposed to be alternative targets to compensate for deficient CFTR-mediated Cl(−) and/or bicarbonate transport in the airways and other affected organs. Theoretically, airway surface liquid hydration could be restored by the inhibition of Na(+) absorption, applying ENaC inhibitors, modulators targeting mutant CFTR such as lumacaftor and ivacaftor or alternative chloride channels modulators. In the present review we will focus on ENaC modulating compounds. Although ENaC is an attractive therapeutic target for rehydration of CF airways, so far no recently developed, adequately acting ENaC inhibitors are available for therapeutic use to restore normal mucociliary clearance. Despite considerable effort, ENaC approaches have been challenging to translate to therapeutic application, mainly because of low or no efficacy, pharmacokinetic problems and off-target side effects such as hyperkalemia. Consequently, clinical trials either have been terminated or completed without introduction of the compounds in therapy (Table 4). It is suggested that transient increases in mean urine Na(+)/K(+) ratios are a first signal of electrolyte imbalances due to renal blockade of ENaC, so that in clinical trials of novel ENaC blockers, intensive measurement of plasma and urine electrolyte levels is required [218].

For example such a compound is BI 1265162, an ENaC inhibitor that was shown to be pre-clinically effective and safe, however with a lack of clinical benefit, such that further drug development has been terminated [219]. Similarly, for the ENaC blocker NVP-QBE170, which was more effective than amiloride in in vivo studies on guinea-pig, rat and sheep without inducing relevant hyperkalaemia [220], no data for clinical studies are available. Inhaled GS-9411 (Table 4) was well tolerated except transient but clinically significant hyperkalaemia, so that this finding also resulted in termination of further clinical drug development. The novel ENaC inhibitor AZD5634 (Table 4) is an effective inhibitor of amiloride-sensitive sodium current in airway cells of healthy persons and in airway cells derived from F508del-homozygous individuals with CF as well as in sheep bronchial epithelial cells. However, in a rat model, nebulized AZD5634 did not show a notable effect and in particular also in a first single-dose study in patients with CF, AZD5634 was ineffective [221]. In phase I studies (Table 4) AZD5634 was safe and well tolerated, because absolute bioavailability of AZD5634 after inhalation was only minimal, so that systemic circulation of the drug was negligible. Inhaled AZD5634 showed an inhibition of nasal ENaC, but did not improve mucociliary clearance. Therefore, further evaluation in multiple dose studies is warranted to explore the therapeutic potential of AZD5634 in CF [222].

Giorgetti et al. [223] used explants from CFTR null mice and tracheobronchial explants from newborn CFTR null piglets to investigate the effects of the ENaC-blockers Compound A, AZD5634 and Parion Compound Ia which are derivatives of amiloride, and the sodium-hydrogen exchanger (NHE) inhibitors AZ1607 and 5-(N,N-Dimethyl)amiloride (DMA) on mucus properties in mouse ileum and mucus bundle transport in piglet airways. Airway mucus bundles were immobile in untreated newborn CFTR null piglets but were detached by the ENaC inhibitor AZD5634. Similarly, the therapeutic drug candidate detached mucus in the mouse ileum, although the mouse ileum lacks ENaC expression. This effect was mimicked by the two NHE inhibiting compounds AZ1607 and DMA [223]. These results suggest that the ENaC inhibitor AZD5634 causes detachment of mucus in the ileum and airway mainly via NHE inhibition, and it is proposed that drug design should focus on NHE instead of ENaC inhibition. However, prior establishment of the type of NHE isoforms which are expressed in human airways is required. As in previously conducted studies where lack or weak efficacy of ENaC inhibitors was observed, weak NHE inhibitory properties of the investigated ENaC blockers could be the reason [223].

Additional approaches to block ENaC are, for example, peptide analogues of SPLUNC1 (short palate, lung, and nasal epithelial clone 1), a multifunctional innate defense protein that is secreted into the airway lumen. SPLUNC1 promotes channel internalization and can inhibit ENaC in vitro [224]. As an allosteric regulator of ENaC, SPLUNC1 dissociates αβγ-ENaC, causes internalisation of αγ-ENaC and generates a new SPLUNC1/β-ENaC complex which remains at the plasma membrane. Additional studies revealed that SPLUNC1 increased NEDD 4-2-dependent ubiquitination of α-ENaC, but not of β- or γ-ENaC [225].

Due to systemic circulation, ENaC inhibitors like amiloride and derivatives may cause unwanted side effects such as hyperkalemia and weight loss. Thus, the aim was to develop small molecules which do not show significant systemic circulation. Such a novel compound is SPX-101 (Table 4). It is a synthetic dodecapeptide mimetic of the natural ENaC inhibitor SPLUNC1 that binds selectively to ENaC and removes ENaC from the membrane by promoting the internalization of α-, β- and γ-subunits. Inhaled SPX-101 increased mucus transport and improved survival in a mouse and in the sheep model for CF [226]. Expression of SPLUNC1 was determined in the sputum from healthy individuals and CF patients. It could be demonstrated that SPX-101 and SPLUNC1 are stable in the sputum in contrast to S18, the ENaC regulatory domain of SPLUNC1. SPX-101 regulated airway surface liquid and increased survival of β-ENaC-transgenic mice after exposure to sputum obtained from CF donors [227]. In a preclinical toxicology assessment nebulized SPX-101 did not show adverse effects in rats and dogs up to the highest preclinical efficacious and expected clinical doses [228]. SPX-101 was well-tolerated across a range of doses and in systemic circulation it was not detected in relevant concentrations [229]. However, clinical trials (Table 4) with the peptide fragment SPX101 did not succeed.

The commonly used macrolides antibiotics are known to have potential immunomodulatory effects. When ENaC-overexpressing human bronchial epithelial cells were treated with the macrolides erythromycin, clarithromycin or azithromycin, a dose-dependent suppression of ENaC function was observed. This led Fujikawa et al. [230] to assume that macrolides might be promising for the treatment of obstructive lung diseases with defective mucociliary clearance.

Inhaled sodium bicarbonate which is used as an adjuvant therapy in CF may have a direct therapeutic effect on the bronchial epithelium [231].

#### 7.2.3. Epigenetic Targeting

Knowledge of the mechanisms of transcriptional regulation of ENaC genes is essential to describe the pathogenic mechanism and the genotype–phenotype relationship in CF. As the *SCNN1A* gene was found to be the most expressed one in different human cell lines in a completely demethylated pattern, while *SCNN1B* and *SCNN1G* genes showed low expression in a pronounced methylated pattern, it seems that modulation of ENaC genes depends on the DNA methylation patterns of specific DNA regions. Dexamethasone treatment stimulated the expression of *SCNN1A*, *SCNN1B* and *SCNN1G* genes, without an evident modulation of the DNA methylation pattern. In nasal brushing, a substantial expression of *SCNN1A, SCNN1B* and *SCNN1G* genes was found despite an apparent methylated pattern. These results suggest an epigenetic controlling mechanism of ENaC function and therefore a possible epigenetic therapeutic approach for CF [232].

Hey et al. [233] used a mouse model of muco-obstructive lung disease (*SCNN1B*-transgenic) in order to identify epigenetically controlled, differentially regulated pathways and transcription factors involved in inflammation. Mucus induces gene expression changes, comparable with those observed in airway macrophages from *SCNN1B*-transgenic mice. Epigenetic reprogramming of airway macrophages is generated by mucostasis, which further causes tissue damage and disease progression. Thus, the researchers [233] suggest these altered airway macrophages as a novel therapeutic approach in patients with muco-obstructive lung diseases.

As the reduction of ENaC activity is a therapeutic option for the treatment of CF, Blacona et al. [234] also considered the novel strategy of epigenetic targeting. They used immortalized human bronchial epithelial cell lines as well as human bronchial primary epithelial cells from healthy individuals and CF patients in order to evaluate the effect of compounds that target the protease-dependent activation of ENaC and the transcriptional activity of its coding genes. While the tested drugs S-adenosyl methionine (SAM) and the serine protease inhibitor camostat are already clinically used to treat other disorders, curcumin is a common dietary compound. In both immortalized and primary cells (with a more pronounced effect on the latter), the inhibition of extracellular peptidases and the epigenetic manipulations reduced ENaC activity. The function of ENaC was evaluated by a fluid absorption assay. Out of the tested genes, *SCNN1B* was proven to be the most suitable target to reduce ENaC activity. Treatment with SAM induced hypermethylation of the *SCNN1B* gene promoter and lowered its expression. It was shown that SAM and curcumin act at the transcription level, while camostat exerted its activity at the protein level. Blacona et al. [234] interpret their findings with the epigenetic modulators SAM and curcumin and the protease inhibitor camostat to the effect that ENaC is downregulated by epigenetic modulation via DNA methylation and chromatin condensation. No further decrease of ENaC function by protease inhibitors was observed as no synergistic or additive actions were found with drug combinations of epigenetic modulators and camostat. Moreover, the application of test compounds either did not affect or even upregulated CFTR expression [234].

#### 7.2.4. Targeting ENaC Genes (Antisense Oligonucleotides, siRNAs)

New approaches to genetic therapies of CF including oligonucleotides that reduce ENaC activity have been recently reviewed in detail [235,236].

Antisense technology enables selective targeting of genes of interest with the advantage of high selectivity, long half-life and thus less frequent administration, and lack of systemic exposure. Antisense oligonucleotides (ASOs) bind specifically to mRNAs, encoding α-, β- and γ-ENaC. Application of ASOs directly into the lung improved disease phenotypes in a CF mouse model [237]. Crosby et al. [238] identified several potent ENaC specific ASOs. The inhaled ASOs decreased ENaC expression by inducing RNase H1-dependent degradation of the targeted *SCNN1A* mRNA. Further, ENaC antisense therapy is expected to benefit CF patients regardless of CFTR mutations [238]. The ASO ENaC-Inhibitor IONIS-ENaCRx is currently being studied in a double-blinded, randomized, placebo-controlled, dose-escalation study to evaluate the safety, tolerability, pharmacokinetics and pharmacodynamics of single and multiple nebulized doses (Table 4). Another ASO, ION-827359 (ENaCCRx), has been studied to evaluate the effect on forced expiratory volume in one second (FEV1) in patients suffering from mild to moderate chronic obstructive disease (COPD) with chronic bronchitis (Table 4). The study was terminated after nine months due to toxicological concerns. Patients who still had received the drug were instructed to stop dosing with a follow-up control period of another 10 weeks.

As ENaC regulation is a promising target, the development of short interfering α-ENaC RNA (α-ENaC siRNA), which mediates silencing of airway ENaC in vitro and *in vivo*, thereby restoring mucociliary function, became a prominent challenge in drug discovery. Translocation of siRNA through the mucus is facilitated by application of siRNA in the form of nanocomplexes. A single dose of siRNA in mouse lung silenced ENaC by approximately 30%, which persisted for at least seven days, and three doses of siRNA even increased silencing to approximately 50%. These data provide clear advantages of nanoparticle-mediated siRNA therapy due to its effectiveness, duration of action, and specific delivery to airway epithelium, which prevents off-target side effects [239]. The investigational drug ARO-ENaC was designed to lower the production of total ENaC. It delivers siRNA to the lungs and thus it was assumed to avoid negative effects on the kidneys. This molecule binds the messenger RNA of α-ENaC and targets it for destruction, preventing α-ENaC proteins to be produced. In healthy sheep ARO-ENaC improved mucus clearance, and in a sheep model of mucus obstruction it preserved lung clearance. However, the phase 1 clinical trial was terminated (Table 4).

#### 7.2.5. Proteolytic Cleavage (CAPs Inhibitors)

Not only does the dysfunction of mutant CFTR generate ENaC hyperactivity, but in addition, an increase in ENaC activity occurs due to the cleavage by proteolytic enzymes which are released from the immune cells in inflamed airways. Channel activating proteases (CAPs) such as neutrophil elastase, prostasin, matriptase and furin were detected in high concentrations at the apical surface of primary airway epithelial cells obtained from patients with CF (F508del homozygotes). The test compound QUB-TL1 inhibited extracellularly located CAPs resulting in reduced ENaC-mediated sodium absorption caused by the internalization of the cleaved γ-ENaC subunit from the cell surface. Further, QUB-TL1-mediated furin inhibition was able to protect against neutrophil elastase-mediated ENaC activation and *Pseudomonas aeruginosa* exotoxin A-induced cell death [240].

#### 7.2.6. Targeting Myeloperoxidase-Derived Oxidants

In CF with excessive neutrophilic inflammation and oxidative stress, the neutrophil enzyme myeloperoxidase produces hypochlorous acid, which is an indicator for worse disease outcomes. Therefore, the pharmacological inhibition of myeloperoxidase in the airways is a challenge to overcome serious complications. Oral administration of the myeloperoxidase inhibitor AZM198 to transgenic β-ENaC overexpressing mice which were infected with *Burkholderia multivorans* prevented hypochlorous acid production in the epithelial lining fluid. Further, the morbidity in mice with lung inflammation was diminished without affecting the clearance of bacteria [241].

### 7.3. Non-Cardiogenic Respiratory Failure

ENaC-related disorders involving pulmonary oedema (Acute Respiratory Distress Syndrome, Acute Lung Injury, High Altitude Pulmonary Oedema and Primary Graft Dysfunction after Lung Transplantation).

The TIP peptides solnatide (AP301), AP318 and AP319 were developed by Apeptico GmbH for various indications involving pulmonary oedema. The underlying rationale is the fact that ENaC plays a critical role in maintenance of optimal alveolar fluid level in the healthy lung which is disturbed in various pathological conditions. Briefly, influx of Na(+) through ENaC in alveolar epithelial cells and subsequent removal by the basolaterally-located ATPase, creates an osmotic gradient along which water follows. Water is ultimately removed from the lungs via the lymphatic or capillary system.

TIP peptides mimicking the lectin-like domain (TIP) of the TNF [242] activate ENaC that is essential for active sodium reabsorption and maintenance of body salt and water homeostasis. The synthetic cyclic peptides, solnatide (AP301), and congeners AP318 and AP319, have been demonstrated in electrophysiological and biochemical experiments to have an activating effect on ENaC. TIP peptides increased the amount of ENaC protein active at the cell surface and increase the open times and Po of the ion channel [47,243,244]. The ENaC activating potential of solnatide and congeners has been shown in vitro using cells expressing ENaC endogenously, such as the H441 cell line, a model of human Na(+) absorptive airway epithelia [245,246], the A549 cell line, derived from human alveolar cell adenoma [243,247] or heterologously, such as HEK-293 cells which have been transfected with ENaC subunits constructed by molecular cloning techniques [47,244,246], and in addition in primary alveolar cells isolated from various animal species [248].

Solnatide has been demonstrated in a variety of animal models of acute lung injury (ALI) to improve alveolar liquid clearance (ALC) in pulmonary permeability oedema of various pathophysiological conditions [246,249,250], and in rats after lung transplantation [251]. Furthermore, solnatide has been shown to reduce pneumolysin-induced pulmonary endothelial hyperpermeability in mice [252] and lysteriolysin-induced hyperpermeability in human pulmonary microvascular endothelial cells [253], suggesting an improvement of the pulmonary barrier function.

The mechanism of solnatide action is complex as it occupies different binding sites to exert its polyvalent function in cells. Binding of TIP peptides to glycosylated residues in the extracellular loop and to an intracellular site of α-ENaC [244,246,254] is essential for activation of ENaC as well as complex formation between ENaC and MARCKS [254]. Molecular docking studies revealed an intracellular site in α-ENaC that is crucial for an increase in Po, but not membrane expression [254]. Two residues, V567 and E568, located in the second transmembrane (TM2) region of α-ENaC, towards the C-terminal end of the TM helix, have been identified as being essential for binding of solnatide [254]. Based on structural information of ENaC [38,39], as well as that of the TIP peptide [243,255] (in regards to the conformational ensemble populated by the peptide in solution [256]), an electrostatic interaction of solnatide with the C-terminal helix of α-ENaC is proposed. In the presence of pneumolysin, binding of solnatide to the C-terminal domain of the α-ENaC stabilizes the ENaC-PIP2-MARCKS complex, which is necessary for Po conformation of ENaC and preserves α-ENaC protein expression by means of blunting the protein kinase C-α pathway [246]. Similarly, solnatide also showed beneficial effects in lysteriolysin O-associated pulmonary oedema formation in mice. The TIP peptide increased lysteriolysin O-mediated decreased ENaC current and expression in H441 cells by preventing the activation of PKC-α, and in addition it restored phospho-Sgk-1 at residue T256 [257]. Human lung microvascular endothelial cells express all three subunits of ENaC as well as acid-sensing ion channel 1a (ASIC1a), which has the capacity to form hybrid non-selective cation channels with α-ENaC [257]. Findings suggest that α-ENaC mediates the protective effect of solnatide in pneumolysin-induced endothelial barrier dysfunction, and that α-ENaC containing non-selective cation channels might be involved in the barrier function of capillary endothelium during pneumonia [258].

To this end, orphan designations have already been granted by the EMA and FDA for solnatide and AP318. The current pharmaceutical formulation of solnatide and AP318 is an aerosolised solution of lyophilisate intended for administration directly to the lungs via oral inhalation. Solnatide has successfully completed a Phase I clinical trial, which demonstrated that orally inhaled AP301 (solnatide) was safe and well-tolerated by all study subjects [259]. It has also been tested in ICU patients in two Phase IIa clinical trials where the peptide as a nebulised spray was introduced directly into the mechanical ventilation system of the patients. The study on alveolar liquid clearance in intensive care patients suffering from ALI (NCT01627613) showed promising results [260], so that two larger phase 2 studies: “Safety and Preliminary Efficacy of Sequential Multiple Ascending Doses of Solnatide to Treat Pulmonary Permeability Oedema in Patients With Moderate-to-severe ARDS“, are currently recruiting (NCT03567577) and ongoing (EudraCT Number: 2017-003855-47) [261]. These phase IIb, randomized, placebo-controlled, double-blind, dose escalation studies will assess the local and systemic safety of solnatide and review the potential efficacy endpoints for a future phase III pivotal trial. With the onset of the COVID-19 pandemic in early 2020 the study design had to be adapted to include COVID-19 patients with acute respiratory distress syndrome (ARDS) (EudraCT Number: 2020-001244-26). Trial specific outcomes regarding pulmonary edema and ventilation parameters did not differ between the COVID-19-ARDS patients and patients with ARDS from different causes, nor did other indicators of (pulmonary) sepsis like oxygenation ratio and required noradrenaline doses [262] (Table 5). The trial EudraCT Nr. 2020-001244-26 contains the Biomarker Sub-Study to assess the role of the coagulation system in the pathogenesis of COVID-19. The incidence of ARDS, its treatment options including solnatide and outcomes also regarding COVID patients have been reviewed [263,264,265,266].

A randomized, placebo-controlled, single-center proof-of-concept pilot-study (NCT02095626), conducted with inhaled solnatide (AP301) or placebo (routine clinical protocol), demonstrated relevant clinical effects of inhaled solnatide on patients with primary graft dysfunction after lung transplantation. Solnatide was administered by nebulizer twice daily for seven days or until extubation. In the solnatide group, only one patient was ventilated at day 4 and no patients thereafter, while in the placebo group, five patients were ventilated on day 4 and two patients on days 5, 6, and 7. The mean increase in the ratio of arterial oxygen partial pressure (PaO_2_ in mmHg) to fractional inspired oxygen (FiO_2_) (Pao_2_/Fio_2_ ratio) was significantly higher in patients treated with solnatide. Moreover, the duration of intubation was shorter in the solnatide group compared with the control group [267].

The effect of solnatide was also investigated in a rat model for high-altitude pulmonary edema (HAPE) in which edema was induced by acute hypobaric hypoxia and exercise. Solnatide reduced the pulmonary edema and increased the expression of the transmembrane protein occludin, which is involved in tight junction formation. Moreover, the gas-blood barrier function was improved during acute hypobaric hypoxia and exercise in rats. These results provide a rationale for the clinical application of solnatide to patients with pulmonary edema and exposure to a high-altitude hypoxic environment [268].

### 7.4. Pseudohypoaldosteronism Type 1 (PHA-1B)

Pseudohypoaldosteronism type 1 (PHA-1) is a hereditary disorder of unresponsiveness to the mineralocorticoid hormone aldosterone, and can be divided into two different forms, showing either a systemic or a renal form of mineralocorticoid resistance. The systemic or generalized form (PHA-1B) with severe symptoms is caused by autosomal recessive loss-of-function mutations in the genes encoding ENaC. Autosomal dominant renal PHA-1A characterized by clinical features improving with age, is caused by loss-of-function mutations in the mineralocorticoid receptor encoding gene NR3C2 [12,269,270].

The rare disease PHA-1B presents in new-borns as life-threatening severe dehydration, hyponatremia and hyperkalemia due to sodium loss involving kidneys, colon, lungs and sweat and salivary glands. Children suffer from pulmonary ailments because reduced sodium dependent liquid absorption results in elevated lung liquid levels. Early diagnosis is critical for survival. The disease shows no improvement with age and patients may require life-long salt supplements, dietary manipulation and potassium chelation treatment. In addition to severe dehydration, vomiting and failure to thrive occurring in the first weeks of life, the clinical picture may be complicated by cardiac arrhythmias, collapse, shock or cardiac arrest. Skin rashes are frequent due to the severe salt loss from sweat glands [145,271].

There is yet no definitive treatment for PHA-1B other than supportive management aimed to reduce sodium wasting and hyperkalemia and to restore water-electrolyte and acid-base balance. Patients suffering from pulmonary oedema are usually treated with a combination of ventilation strategies, administration of β2-adrenergic receptor agonists and diuretics to stimulate lung fluid clearance. Treatment of PHA-1B is complicated and rendered problematical by its systemic, multi-organ nature, especially in new-borns [271].

The disease is caused by mutations in the genes *SCNN1A* [135,145,152,272,273,274,275,276,277,278,279,280,281,282,283,284,285,286,287,288], *SCNN1B* [135,146,151,272,273,274,275,284,285,289,290,291,292,293,294,295,296,297] and *SCNN1G* [283,286,290,298,299,300] encoding for the ENaC subunits (Appendix A [38,301]). Mutations of ENaC causing PHA-1B result in reduced or complete loss of ENaC function, depending on the type of mutation. Search for mutations in databases such as ClinVar is not always reliable, because many mutations in patients are not defined and/or not reported. An exhaustive search of the literature has revealed about 70 different mutations known to cause PHA-1B, either in the homozygous or in the compound heterozygous state. These mutations are in the majority of cases a substitution, addition or deletion of single nucleotide bases in one of the three genes *SCNN1A*, *SCNN1B* or *SCNN1G* encoding for α-, β- and γ-ENaC subunits, respectively. No mutations causing PHA-1B have yet been found in the *SCNN1D* gene encoding for δ-ENaC (Table 1).

Using site directed mutagenesis, individual PHA-1B mutations observed in patients, have been reproduced. Human wild type and PHA1-B mutant ENaC subunits have been expressed heterologously to test the ability of the TIP peptides, solnatide and AP318, to activate reduced amiloride-sensitive sodium currents of the loss-of-function PHA-1B mutant ENaC. About 20 mutants have been tested electrophysiologically in the presence of solnatide and 10 of these in the presence of AP318. In all these cases, solnatide or AP318 potentiated the amiloride-sensitive sodium current to levels normally observed for wild type control ENaC, i.e., to physiological levels. Furthermore, both solnatide and AP318 affected the amount of mutant ENaC expressed at the cell surface to different extents in different PHA-1B mutants [302,303]. Solnatide and AP318, as aerosolized solutions, can be delivered directly to their target, ENaC, which are located on the apical side of the alveolar epithelium. Application is primarily aimed at treating the respiratory symptoms and complications in PHA1B patients. Orphan drug designation has been granted by the the EMA for solnatide and AP318, and by the EMA and FDA for solnatide, for treatment of PHA-1B.

In Figure 5 a schematic overview of potential drug targets for ENaC modulation is illustrated as discussed in the present review article.

### 7.5. Inflammatory Diseases

#### Arthropathies, Nephritis

Gram-negative bacterial lipopolysaccharide (LPS) increases the susceptibility of cells to inflammatory diseases and septic syndrome. Apical application of LPS activated Ca(2+)-activated Cl(−) channels, as well as ENaC, thereby disrupt the epithelial barrier integrity which results in the depolarization of the apical membrane. LPS-impaired epithelial barrier became susceptible to secondary bacterial infections, which could be antagonized by the inhibition of Ca(2+)-activated chloride channels and ENaC [304].

The dichotomous role of TNF in pulmonary barrier function and ALI has been extensively reviewed by Lucas et al. [305], with emphasis on the divergent actions of TNF via its receptor binding sites versus its lectin-like domain in the lungs and the role of ENaC in disease conditions.

Solnatide has been shown to resolve edema during inflammation [252,253,257] and to improve barrier function in the capillary endothelium during pneumonia [258]. Thus, the effect of intraperitoneally applied and glomerularly delivered solnatide was investigated in a mouse model, in which the initial stage of nephrotoxic serum-induced nephritis mimics antibody-mediated human glomerulonephritis. The reduction of inflammation, proteinuria and blood urea nitrogen was blocked by the cyclooxygenase inhibitor indomethacin, indicating the involvement of prostaglandins. In severe nephrotoxic serum-induced nephritis, targeted application of solnatide even reduced mortality of the mice. Effects were attributed to the activation of ENaC in glomerular endothelial cells upon binding of solnatide to the α-subunit. In the later autologous phase of nephrotoxic serum-induced nephritis, solnatide blunted the infiltration of pro-inflammatory Th17 cells [306].

Because the TNF is involved in joint destruction, TNF-inhibitors are commonly used for the treatment of arthropathies, although not all patients experience a beneficial effect but an additional increased susceptibility to infections. Hence, TNF-receptor-independent activities of TNF, which are mediated by the TNF-derived solnatide peptide, were studied in an immune-mediated arthritis mouse model. Although chondrocytes and osteoblasts were shown to express ENaC receptors, it is likely that endothelial cells and type II macrophage-like synoviocytes are the primary targets of solnatide following intraarticular injection and intravenous administration. Solnatide binds to ENaC channels in endothelial cells and down-regulates the release of pro-inflammatory chemokines and cytokines, which causes reduced cell recruitment and subsequently the reduced release of additional inflammatory mediators, resulting in diminished inflammation and local edemas [307]. This is in accordance with data by Madaio et al. [306] who demonstrated that depletion of the α-ENaC subunit blunted the protective effect of solnatide in TNF-induced inflammation in glomerular endothelial cells. Thus, it is proposed that solnatide exerts its anti-inflammatory effects in the arthritis mouse model via an interaction between the lectin-like domain of TNF, mimicked by solnatide, and ENaC. This corroborates with the finding that N-acetylcysteine, which blocks lectin-like activity of TNF, worsens the symptoms [307].

### 7.6. Concluding Remarks

Considerable progress has been achieved in the elucidation of regulation, expression, structure and physiological role of ENaC and its involvement in pathophysiological conditions, leading some investigators to propose ENaC targeting as a promising new therapeutic approach. Consequently, numerous compounds have been investigated in vitro and some also in appropriate animal models with partly encouraging results. But only a few of these compounds were studied in clinical settings with a rather low successful output so far, indicated by the termination of the trial or no further proof in phase 2 clinical studies. An exception is the orphan drug solnatide, representative of newly developed drugs, which is currently being tested in phase 2 clinical trials in the setting of ARDS (NCT03567577, EudraCT Number: 2017-003855-47, EudraCT Number: 2020-001244-26). The NOX1/ NOX4 inhibitor setanaxib, which indirectly affects ENaC, is tested in clinical phase 2 and 3 trials for therapy of primary biliary cholangitis, liver stiffness and squamous cell carcinoma of the head and neck. The established ENaC blocker amiloride is mainly used as an add-on drug in the therapy of resistant HTN and is being studied in ongoing clinical phase 3 and 4 trials for special applications. ENaC is part of a complex network of interacting biochemical pathways and as such is implicated in several disease states brought about by their dysfunction. Therefore, on the one hand ENaC is a promising target in drug discovery for direct or indirect modulation of the channel with the aim of treating disorders of different origins. But on the other hand, owing to ENaC’s properties, the drug development process is challenging, mainly due to pharmacokinetic characteristics and the off-target side effects in tissues and organs other than those anticipated.

Autosomal recessive pseudohypoaldosteronism type 1:PHA1B, OMIM:264350 (https://www.omim.org/entry/264350) accessed on 27 March 2023

Global Variome shared LOVD database [153]:https://databases.lovd.nl/shared/genes/SCNN1A accessed on 27 March 2023https://databases.lovd.nl/shared/genes/SCNN1B accessed on 27 March 2023https://databases.lovd.nl/shared/genes/SCNN1G accessed on 27 March 2023

Liddle Syndrome (LS):LIDLS, OMIM:177200 (https://www.omim.org/entry/177200) accessed on 27 March 2023

Cystic fibrosis (Mucoviscidosis):CF, OMIM:219700 https://www.omim.org/entry/219700 accessed on 27 March 2023

Bronchiectasis (Cystic fibrosis-like syndrome):BESC1, OMIM:211400 https://www.omim.org/entry/211400 accessed on 27 March 2023BESC2, OMIM:613021 https://www.omim.org/entry/613021 accessed on 27 March 2023BESC3, OMIM:613071 https://www.omim.org/entry/613071 accessed on 27 March 2023

Clinical trials:https://clinicaltrials.gov/amiloride accessed on 27 March 2023https://clinicaltrials.gov/ENaC accessed on 27 March 2023https://clinicaltrials.gov/setanaxib, https://clinicaltrials.gov/GKT137831 accessed on 27 March 2023https://clinicaltrials.gov/solnatide, https://clinicaltrials.gov/AP301 accessed on 27 March 2023https://eudract.ema.europa.eu/ENaC accessed on 27 March 2023https://eudract.ema.europa.eu/solnatide accessed on 27 March 2023https://eudract.ema.europa.eu/AP301 accessed on 27 March 2023

## Figures and Tables

**Figure 2 ijms-24-07775-f002:**
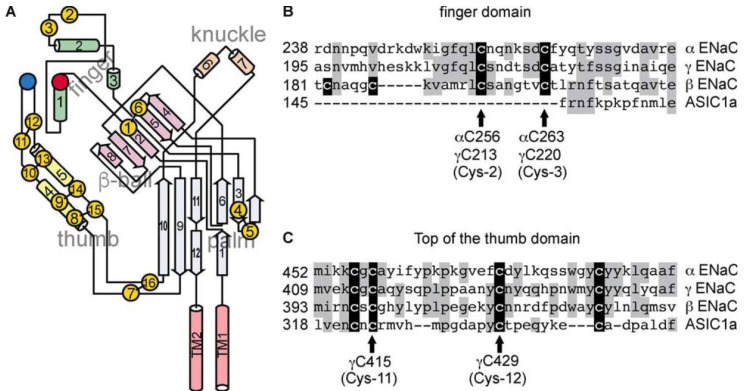
ENaC subunit cysteines (mouse ENaC). (**A**) A Schematic of an ENaC subunit, adapted from [5]. Cysteine residues are represented by yellow circles, and disulfide bonds observed in the ASIC1 structure are represented by black bars. Conserved aspartic acids αAsp-171 and γAsp-115 are represented by a red circle, and αTyr-474 is represented by a blue circle. (**B**) *B* Sequence alignment of finger domain segment hosting conserved cysteines in ENaC subunits. (**C**) *C* Sequence alignment of the thumb domain residues proximal to the finger domain in the ASIC1 structure. Identities are shaded gray. Cysteines are shaded black. Reproduced from [83] (Figure 1); Copyright © 2023 by The American Society for Biochemistry and Molecular Biology, Inc.; published by American Society for Biochemistry and Molecular Biology, Baltimore, MD, USA.

**Figure 3 ijms-24-07775-f003:**
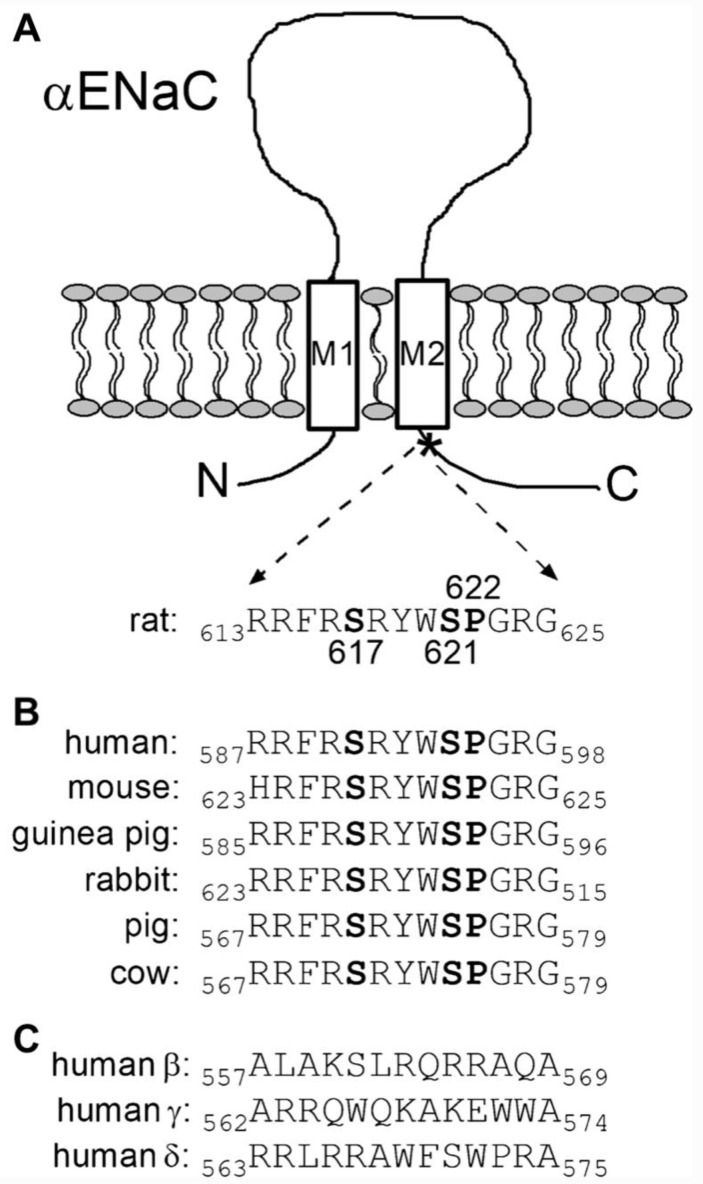
Potential sites of phosphorylation in α-ENaC C-terminal region. Two serine residues and one proline residue are highly conserved in a C-terminal region of α-ENaC close to the second transmembrane domain. (**A**) Schematic representation of α-ENaC illustrating the extracellular loop, two transmembrane domains (M1 and M2) and intracellular N- and C-termini. The amino acid sequence of rat α-ENaC (residues 613–625) corresponds to the C-terminal region indicated by a star (*) and contains the serine residues 617 (S617) and 621 (S621) and the proline residue 622 (P622) highlighted in bold. (**B**) Amino acid sequence alignment of this highly conserved C-terminal region from several mammalian α-ENaC subunits. The residues homologous to S617, S621 and P622 in rat αENaC are highlighted in bold. (**C**) Amino acid sequence alignment of homologous C-terminal regions from human β-, γ- and δ-ENaC subunits, showing a lack of potential phosphorylation sites. Adapted from [110] (Figure 1); Copyright © The Author(s) 2022, corrected publication 2022; published by Springer Nature.

**Figure 4 ijms-24-07775-f004:**
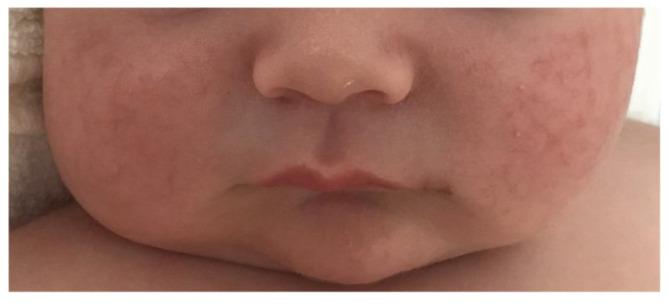
PHA-1B patient with *miliaria rubra* on the face. Reproduced from [145] (Figure 1); Copyright © The Author(s) 2021; published by BioMed Central, London.

**Figure 5 ijms-24-07775-f005:**
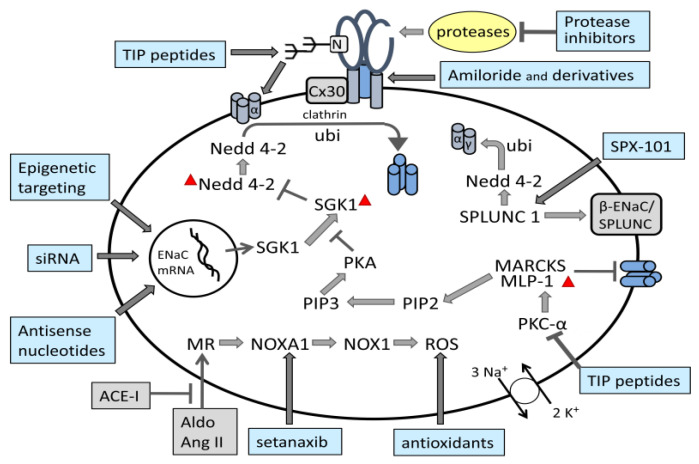
Simplified scheme of ENaC regulation and targets for ENaC modulation. *Abbreviations:* ACE-I (angiotensin-converting enzyme inhibitors), Aldo (aldosterone), AngII (angiotensin II), Cx30 (connexin 30), MLP-1 (myristoylated alanine-rich C kinase (MARCKS)-like protein 1), MR (mineralcorticoid receptor), N (asparagine), Nedd 4-2 (neural precursor cell expressed developmentally down-regulated 4-2), NOXA1 (reduced form of nicotinamide adenine dinucleotide phosphate (NADPH) oxidase activator 1), PIP2 (phosphatidylinositol 4,5-*bis*phosphate), PIP3 (phosphatidylinositol 4,5-triphosphate), PKC-α (proteinkinase C isoform α), siRNA (short interfering ribonucleic acid), SPLUNC1 (short palate, lung, and nasal epithelial clone 1), ROS (reactive oxygene species), SGK1 (serum and glucocorticoid-regulated kinase 1), ub (ubiquitination). Red triangles indicate phosphorylation.

**Table 1 ijms-24-07775-t001:** Types of mutation reported in the literature to cause autosomal recessive PHA-1 (PHA-1B) and distribution amongst ENaC subunit genes (Appendix A).

Type of Mutation *	Gene		
	*SCNN1A*	*SCNN1B*	*SCNN1G*
Missense	13	2	1
Nonsense (stop)	6	3	1
Frameshift	14	10	2
In-frame deletion	1		1
Deletion, insertion or substitution of nucleotides in intron or splice junction	7	5	2
Large upstream deletion in non-coding region		1	
TOTAL per gene	41	21	7

* For details of mutation and publication see Appendix A.

**Table 2 ijms-24-07775-t002:** Clinical trials with amiloride (https://clinicaltrials.gov/, accessed on 27 March 2023).

Study Title/Study Number	Intervention	Trial Phase/ Recruitment Status	Disease/ Condition
Data Analysis for Drug Repurposing for Effective Alzheimer’s Medicines (DREAM)-Amiloride vs Triamterene NCT05125237	Amiloride Triamterene	Retrospective 2021–2023 Active, Not recruiting	Hypertension
The effect of Amiloride and Spironolactone in Patients With Hypertension (hass) NCT01388088	Spironolactone Amiloride placebo	Phase 4 2010–2012 completed	Hypertension
Bioequivalence Study Between GSK3542503 Hydrochlorothiazide + Amiloride Hydrochloride 50 mg: 5 mg Tablets and Reference Product in Healthy Adult Participants Under Fasting Conditions NCT03031496	GSK3542503 (HCTZ 50 mg/ Amiloride HCl 5 mg tablets) Moduretic (HCTZ 50 mg/ Amiloride HCl 5 mg tablets)	Phase 1 2017 completed	Hypertension
Comparison of Spironolactone and Amiloride on Home Blood Pressure in Resistant Hypertension NCT04331691	Spironolactone amiloride	Phase 4 2020–2024 recruiting	Resistant hypertension
Diuretics and Angiotensin-Receptor Blocker Agents in Patients With Stage I Hypertension NCT00971165	Losartan Chlorthalidone plus amiloride	Phase 3 2010–2014 completed	Hypertension Cardiovascular Disease
Amiloride for Resistant Hypertension NCT02122731	Amiloride added to triple antihypertensive therapy	Phase 4 2010–2012 completed	Hypertension Type 2 Diabetes Microalbuminuria
Epithelial Sodium Channel (ENaC) as a Novel Mechanism for Hypertension and Volume Expansion in Type 2 Diabetes NCT01804777	Amiloride hydrochlorothiazide	Early phase 1 2013–2014 terminated	Proteinuria Hypertension Type II Diabetes
Amiloride in Nephrotic Syndrome (AMILOR) NCT05079789	amiloride 5 mg furosemide 40 mg	Phase 3 2020–2023 recruiting	Nephrotic Syndrome Edema Na(+) Retention
Safety and Efficacy of Furosemide 40 mg + Amiloride Hydrochloride 10 mg to Reduct Edema NCT01210365	fixed combination of furosemide and amiloride (10 mg) furosemide (40 mg)	Phase 3 2011–2013 terminated	Congestive heart failure Oedema
Treatment of Vascular Stiffness in ADPKD NCT05228574	NaCl Placebo Amiloride 5 mg	Phase 4 2022–2024 recruiting	Autosomal Dominant Polycystic Kidney
ENaC Blockade and Arterial Stiffness NCT03837626	Placebo-Cap	Phase 2 2019–2024 recruiting	overweight and obesity
	Amiloride Pill	Phase 3 2019–2024 recruiting	Insulin resistance

**Table 3 ijms-24-07775-t003:** Clinical trials with setanaxib (GKT137831). (https://clinicaltrials.gov/, https://eudract.ema.europa.eu/, accessed on 27 March 2023).

Study Title/ Study Number	Intervention	Trial Phase/ Recruitment Status	Disease/ Condition
Study to Evaluate the Pharmacokinetics and Drug-Drug Interactions of Setanaxib in Healthy Adult Male and Female Subjects NCT04327089	Setanaxib	Phase 1 2020–2022 completed	Pharmacokinetics
A Trial of Setanaxib in Patients With Primary Biliary Cholangitis (PBC) and Liver Stiffness NCT05014672 EudraCT Nr. 2021-001810-13	Placebo Setanaxib (GKT137831)	Phase 2 2021–2023 recruiting Phase 3	Primary Biliary Choleangitis Liver Stiffness
A Study of Setanaxib Co-Administered With Pembrolizumab in Patients With Recurrent or Metastatic Squamous Cell Carcinoma of the Head and Neck (SCCHN) NCT05323656	Setanaxib Pembrolizumab Placebo	Phase 2 2022–2023 recruiting	Squamous Cell Carcinoma of Head and Neck
Study to Assess Safety & Efficacy of GKT137831 in Patients With Primary Biliary Cholangitis Receiving Ursodiol. NCT03226067	Placebo Setanaxib (GKT137831)	Phase 2 2017–2022 completed	Primary Biliary Cirrhosis

**Table 4 ijms-24-07775-t004:** Clinical trials with experimental compounds targeting ENaC. (https://clinicaltrials.gov/, https://eudract.ema.europa.eu/, accessed on 27 March 2023).

Test Compound	Study Title/Study Number	Intervention	Trial Phase/ Recruitment Status	Disease/ Condition
**ENaC inhibitors**				
VX-371	Clearing Lungs with ENaC Inhibition in Primary Ciliary Dyskinesia (CLEAN-PCD) NCT02871778	VX-371 Hypertonic Saline Placebo VX-371 + HS Ivacaftor	Phase 2 2016–2021 completed	Primary Ciliary Dyskinesia
VX-371	Safety and Efficacy of VX-371 Solution for Inhalation With and Without Oral Ivacaftor in Subjects With Primary Ciliary Dyskinesia EudraCT2015-004917-26	VX-371 (inhalation) Ivacaftor (oral) Placebo	Phase 2a 2016–2018 completed	Primary Ciliary Dyskinesia
P-1037	Clearing Lungs With Epithelial Sodium Channel (ENaC) Inhibition in Cystic Fibrosis (CF) (CLEAN-CF) NCT02343445	P-1037 Hypertonic Saline Saline	phase 2 2015–2021 completed	Cystic fibrosis
ETD001	A First in Human Study to Evaluate the Safety, Tolerability and Pharmacokinetics of Single and Multiple Ascending Doses of Inhaled ETD001 NCT04926701	inhaled, single dose, and multiple once/twice daily ETD001 Placebo	Phase 1 2021–2022 completed	Cystic fibrosis
GS-9411	A Phase 1 Trial to Assess the Safety, Tolerability, and Pharmacokinetics of GS 9411 in Subjects With Cystic Fibrosis (CF) NCT01025713	GS-9411 Placebo	Phase 1 2009–2010 withdrawn	Cystic fibrosis Mucociliary Clearance
GS-9411	Trial to Assess the Safety, Tolerability and Pharmacokinetics of GS-9411 in Healthy Male Volunteers NCT00800579	GS-9411 Placebo	Phase 1 2008–2009 completed	Cystic fibrosis
GS-9411	A Trial to Assess the Safety, Tolerability, and Pharmacokinetics of GS-9411 in Healthy Male Volunteers NCT00951522	GS-9411 Placebo	Phase 1 2009	Cystic fibrosis Mucociliary Clearance Airway Hydration
AZD5634	To Assess the Safety, Tolerability and Pharmacokinetics of AZD5634 Following Inhaled and Intravenous (IV)Dose Administration NCT02679729	AZD5634 for inhalation AZD5634 for infusion Placebo	Phase 1 2016 completed	Cystic fibrosis
AZD5634	A Study to Assess the Effect of AZD5634 on Mucociliary Clearance, Safety, Tolerability and Pharmacokinetic Parameters in Patients With Cystic Fibrosis NCT02950805	Placebo AZD5634	Phase 1 2017–2018 completed	Pulmonary/Respiratory Diseases
BI 1265162	A 4-week Study to Test Different Doses of BI 1265162 in Adolescents and Adults With Cystic Fibrosis Using the Respimat^®^ Inhaler NCT04059094	BI 1265162 Placebo	Phase 2 2020 Terminated	Cystic fibrosis
**Analogues of SPLUNC1**				
SPX-101	A Safety Study of SPX-101 Inhalation Solution in Subjects With Cystic Fibrosis NCT03056989	SPX-101	Phase 1 2017 completed	Cystic fibrosis
SPX-101	An Efficacy and Safety Study of SPX-101 Inhalation Solution in Subjects With Cystic Fibrosis NCT03229252	Placebo SPX-101	Phase 2 2017–2019 completed	Cystic fibrosis
**Antisense oligonucleotides**				
IONIS-ENaCRx	A Phase 1/2a Study to Assess the Safety, Tolerability, Pharmacokinetics and Pharmacodynamics of Single and Multiple Doses of IONIS-ENaCRx in Healthy Volunteers and Patients With Cystic Fibrosis NCT 03647228	IONIS-ENaCRx Placebo Ascending single and multiple doses inhaled or nebulized	Phase 1 2018–2020 Completed	Healthy Subjects Cystic fibrosis
ION-827359 Antisense Inhibitor of ENaC (ENaCCRx)	A Double-Blind, Placebo-Controlled, Phase 2a Study to Assess the Safety, Tolerability, and Efficacy of ION-827359 in Patients with Mild to Moderate COPD with Chronic Bronchitis EudraCT2020-000210-15 NCT04441788	Placebo ION-827359 Oral inhalation of single-dose once every week up to 13 weeks	Phase 2a 2020–2021 terminated	COPD with Chronic Bronchitis
**Short interfering α-ENaC RNA**				
ARO-ENaC	Study of ARO-ENaC in Healthy Volunteers and in Patients With Cystic Fibrosis NCT04375514	ARO-ENaC Placebo	Phase 1 2020–2022 terminated	Cystic Fibrosis Pulmonary

**Table 5 ijms-24-07775-t005:** Clinical trials with solnatide (AP301). (https://clinicaltrials.gov/, https://eudract.ema.europa.eu/, accessed on 27 March 2023).

Study Title/ Number	Intervention	Trial Phase/ Recruitment Status	Disease/ Condition
Study in Intensive Care Patients to Investigate the Clinical Effect of Repetitive Orally Inhaled Doses of AP301 on Alveolar Liquid Clearance in Acute Lung Injury NCT01627613 EudraCT Nr. 2012-001863-64	AP301 Saline Solution	Phase 2 2012–2019 completed	Acute Lung Injury (ALI)
Study in Intensive Care Patients Regarding the Effect of Inhaled AP301 After Primary Graft Dysfunction After Lung Transplantation NCT02095626 EudraCT Nr. 2013-000716-21	AP301 Saline Solution	Phase 2 2014–2019 completed	Primary Graft Dysfunction
Safety and Preliminary Efficacy of Sequential Multiple Ascending Doses of Solnatide to Treat Pulmonary Permeability Oedema in Patients With Moderate-to-severe ARDS NCT03567577	Solnatide 25 mg powder (solution for inhalation) 0.9% Saline Solution	Phase 2 2018–2022 recruiting	ARDS
COVID-19: Efficacy of solnatide to treat pulmonary permeability oedema in SARS-CoV-2 positive patients with moderate-to-severe ARDS—a pilot-trial. EudraCT Nr. 2020-001244-26	Solnatide Powder for nebuliser suspension (inhalation) Placebo	Phase 2 2020–2021 Prematurely ended	Oedema in SARS-Co-2 positive patients with ARDS

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
