# Peer review of "The Epithelial Sodium Channel—An Underestimated Drug Target"

_ijms, 2023, doi:10.3390/ijms24097775_

Round 1
Reviewer 1 Report
The review "The epithelial sodium channel - an underestimated drug target" is a well presented and very nice review of the ENaC family and their modifiers. The authors showed the ENaC regulation, the importance of these channels for many cell functions and the main pathologies related to them. Further, their structural functioning is presented as well as the clinical trials concerning these channels.
The figures and tables are clear and helpful.
The review will contribute to better understanding the role of ENaC channels as well as to draw attention to these channels and to the need to develop new modulators.
Author Response
Dear reviewer,
thank you very much for reviewing the manuscript
Reviewer 2 Report
The authors have brought together a large amount of information to provide an extensive review on ENaC structure, regulation and involvement in several pathologies. They then review development and testing of pharmaceuticals to target ENaC in several pathologies, including tabulation of relevant clinical trials and their outcomes if known. Due to ENaC being active in multiple cell types and tissues the authors are balanced in their writing indicating that potential off-target effects of any drugs need to be carefully studied.
Please note the symbol font has not been copied over correctly - ENaC subunit names are spelt out.
Major comments
1. Page 3, line 103, deltaENaC has been described in some rodents, just not mice or rats, see https://pubmed.ncbi.nlm.nih.gov/34491346/
2. Page 9, line 333 (and multiple other instances), please change C-termini /N-termini to C/N-terminal regions, particularly when referring to the PY motif which is located many amino acids internal to the C-termini of the ENaC subunits.
3. Page 24, line 932: Piezo is the best understood mechanosensitive ion channel – suggest add Piezo into this sentence.
4. Page 28, Table 3 is focused on cystic fibrosis when the title of this section is ‘Putative targets and ENaC-modulating compounds for treatment of cardiovascular disease’. This table seems to be have been placed in an inappropriate section, as the text regarding information in this table (e.g. how SPLUNC1 fits into the story, if the antisense oligonucleotides target ENaC, and the outcomes of these studies) is found in section 7.2. This reviewer asks that this table be moved to section 7.2 where the text related to this table is located.
5. Page 34, section 7.2.1, the authors focus on the possibility of targeting ENaC overactivity in CF, however it is important to mention that many CF patients are now being successfully treated with Vertex compounds that directly increase trafficking of CFTR to the cell surface or increase open probability. Combinations of these compounds e.g. Trikafta are making a huge difference for this community. I suggest the studies targeting CFTR are more clearly summarised early on in this section, and include a clear explanation about why ENaC inhibition should also be targeted in CF and related lung conditions.
Minor comments
1. Add solnatide to graphical abstract as it is listed as a keyword.
2. Reference 25 ‘ΑENaC’ should be ‘alphaENaC’, also in several other references e.g. 42, 50, 84.
3. References 50, 108, 130, 131 should have USA added after Proc Natl Acad Sci
4. Page 3, line 90, suggest alter ‘clues as the’ to ‘clues as to the’
5. Page 4, line 161, reference 21 is an original research article, reference 20 is a review – should reference 20 be swopped in for reference 21?
6. Page 5, line 196, suggest add ‘and’ to read ‘…[69], and transmembrane…’
7. Figure 1 legend: there is only one cysteine protease and only one metalloprotease, therefore the sentence should read ‘…cysteine protease in orange, and metalloprotease in brown.
8. Page 9, line 337, all the information from this line to the end of the paragraph is from the Frindt et al. 2020 paper, therefore suggest Frindt et al (2020) reported that ubiquitinated alphaENaC…..’
9. Page 9, line 341 I believe that this paper reported that the 566X mutation was made in the betaENaC gene (not the gammaENaC gene). This is the original mutation reported by Shimkets et al in 1995.
10. Page 9, line 356 – suggest the original reference(s) are cited rather than the review [48].
11. Page 9, line 358, I believe that the endocytosis motif was first predicted by Snyder et al: https://pubmed.ncbi.nlm.nih.gov/8521520/
12. Page 11, line 424, suggest add in figure legend ‘..in bold, showing a lack of potential phosphorylation sites.’
13. Page 12, line 446, suggest add ‘..open state, reviewed in [48].’
14. Page 12, line 456, add a space between protein and kinase.
15. Page 13, you could add that SGK1 phosphorylates Nedd-2.
16. Page 13, line 498, enzym should be enzyme. Also page 21, line 810.
17. Page 14, line 527, recommend reference 128 is changed to reference 129 as this is the original research article showing gain of function mutations in both beta- and gammaENaC genes.
18. Page 14, line 529, suggest add Snyder et al. 1995, https://pubmed.ncbi.nlm.nih.gov/8521520/
19. Page 14, line 534, reference 134 does not provide any ubiquitination data. Suggest using this reference: https://pubmed.ncbi.nlm.nih.gov/17502380/
20. Page 17, line 648, suggest alter to read ‘..caused by excessive salt…’
21. Page 19, line 709, suggest mention all ENaC subunits as you have mentioned the alpha,beta,gammaENaC subunits previously. There is also evidence that variants of the dENaC subunit are associated with hypertension: https://pubmed.ncbi.nlm.nih.gov/36193739/
22. Page 21, line 821, please clarify the function of benzbromarone.
23. Page 22, line 834 – please clarify the intended meaning of this sentence. Should it read: ‘Besides being implicated in HTN, cardiovascular disease, and kidney disease, excessive salt intake….’
24. Page 24, line 910 – betaENaC isn’t a DEG protein – but betaENaC is a member of the DEG/ENaC family. Suggest rephrase sentence to clarify this.
25. Page 24, line 922, correct ENa to ENaC.
26. Page 24, line 922, presumably use of mouse (no deltaENaC) versus human studies influence these studies?
27. Page 31, line 1056, please correct ‘thoroughly’
28. Reference 231, I think the first author’s surname is Boyd, and Christopher is their preferred first name, so use: Boyd, AC in the reference list.
Author Response
Dear reviewer,
thank you very much for reviewing the manuscript so carefully and the very useful suggestions.
Replies to the comments are attached.
